# COSMOS-UK: National soil moisture and hydrometeorology data for environmental science research

Hollie M. Cooper[1], Emma Bennett[1], James Blake[1], Eleanor Blyth[1], David Boorman[1], Elizabeth Cooper[1], Jonathan Evans[1], Matthew Fry[1], Alan Jenkins[1], Ross Morrison[1], Daniel Rylett[1], Simon Stanley[1], Magdalena Szczykulska[1], Emily Trill[1], Vasileios Antoniou[1], Anne Askquith-Ellis[1], Lucy Ball[1], Milo Brooks[2], Michael A. Clarke[3], Nicholas Cowan[4], Alexander Cumming[1], Philip Farrand[1], Olivia Hitt[a], William Lord[1], Peter Scarlett[1], Oliver Swain[1], Jenna Thornton[1,b], Alan Warwick[1], Ben Winterbourn[1,c]

[1]UK Centre for Ecology & Hydrology, Wallingford, OX10 8BB, United Kingdom
[2]UK Centre for Ecology & Hydrology, Environment Centre Wales, Bangor, Gwynedd, LL57 2UW, United Kingdom
[3]UK Centre for Ecology & Hydrology, Lancaster Environment Centre, Bailrigg, Lancaster, LA1 4AP, United Kingdom
[4]UK Centre for Ecology & Hydrology, Bush Estate, Penicuik, Midlothian, EH26 0QB, United Kingdom
[a]formerly at: UK Centre for Ecology & Hydrology, Wallingford, OX10 8BB, United Kingdom
[b]now at: UK Met Office, Cardington Airfield, Shortstown, Bedford, MK42 0SY, United Kingdom
[c]now at: School of Ocean Sciences, Bangor University, Bangor, Gwynedd, LL59 5AB, United Kingdom

*Correspondence to*: Hollie M. Cooper (holcoo@ceh.ac.uk)

**Abstract.** The COSMOS-UK observation network has been providing field scale soil moisture and hydrometeorological measurements across the UK since 2013. At the time of publication a total of 51 COSMOS-UK sites have been established, each delivering high temporal resolution data in near-real time. Each site utilises a cosmic-ray neutron sensor, which counts epithermal neutrons at the land surface. These measurements are used to derive field scale near-surface soil water content, which can provide unique insight for science, industry, and agriculture by filling a scale gap between localised point soil moisture and large-scale satellite soil moisture datasets. Additional soil physics and meteorological measurements are made by the COSMOS-UK network including precipitation, air temperature, relative humidity, barometric pressure, soil heat flux, wind speed and direction, and components of incoming and outgoing radiation. These near-real time observational data can be used to improve the performance of hydrological models, validate remote sensing products, improve hydro-meteorological forecasting and underpin applications across a range of other scientific fields. The most recent version of the COSMOS-UK dataset is publically available at https://doi.org/10.5285/b5c190e4-e35d-40ea-8fbe-598da03a1185 (Stanley et al., 2021).

## 1 Introduction

Soil moisture plays a crucial role in a range of biogeophysical and biogeochemical land surface processes (Moene and van Dam, 2014; Seneviratne et al., 2010). These processes include the transport of energy and matter via evapotranspiration, drainage, run-off, infiltration and plant photosynthesis, and controlling aerobicity of soils. Since 2013 the UK Centre for Ecology & Hydrology (UKCEH) has established the world's most spatially dense national network of innovative cosmic-ray neutron sensors (CRNSs) to monitor soil moisture across the UK. The Cosmic-ray Soil Moisture Observing System for the UK (COSMOS-UK) delivers field scale near-surface soil water content for around 50 sites in near-real time (https:\\cosmos.ceh.ac.uk). The field scale measurement footprint of these soil moisture observations, collocated with hydrometeorological measurements, is directly relevant to Land Surface Models (LSMs) and Earth Observation (EO) data products. COSMOS-UK therefore aims to transform hydrological and land surface modelling and monitoring, enabling and supporting a range of applications across science and industry.

Whilst the UK has a long history and well-established tradition of monitoring meteorological and hydrometeorological variables, namely precipitation, temperature and river flow, soil moisture has until recently been difficult to measure in a cost effective way and at a scale appropriate to many applications. Real-time soil moisture information is crucial in understanding the susceptibility of rainfall to cause flooding, the need for irrigation, the likelihood of landslip, and the suitability of undertaking agricultural activities. Additionally, knowledge of the soil moisture regime informs all land-use planning, the need for drainage, water resource development, flood forecasting, drought management, and agricultural development. High frequency soil moisture measurements are also crucial to the development of process based models which replicate soil and microbial processes in soils, which significantly influence greenhouse gas (GHG) emissions and the nitrogen cycle in natural and agricultural systems (Oertel et al., 2016). With the absence of appropriate sensor technology, most notably due to the gap in spatial scale between small-area sensors and large-area remote sensing, soil moisture information has historically been

estimated by hydrological and land-surface models. The development and use of the CRNS provides appropriate scale data to

enable model application, calibration and testing as well as providing near-real time data of local relevance.

COSMOS-UK fills a critical gap in UK hydrological monitoring by utilising CRNS to monitor field scale soil moisture (see the UK Water Resources Portal, https://eip.ceh.ac.uk/hydrology/water-resources/). At each COSMOS-UK site the CRNS sits above ground, autonomously counting epithermal neutrons for near-real time processing at UKCEH. The instrument has a measurement footprint of approximately 12 hectares, and can measure to a depth of approximately 80 cm depending on local

conditions (see Sect. 3.1 for details). This therefore fills the scale gap between buried point soil sensor measurements and very near surface soil data captured in EO soil moisture products. CRNS data are being used across the globe, including from networks in the United States (Zreda et al., 2012), Australia (Hawdon et al., 2014), Germany (Baatz et al., 2014; Fersch et al., 2020), Kenya and India (Montzka et al., 2017; Upadhyaya et al., 2021). COSMOS-UK aims to support science, industry and agriculture by providing reliable, accurate and timely soil moisture information for the UK.

This paper introduces the COSMOS-UK network and the data available for use. Current instrumentation and protocols are described in Sect. 2. Section 3 outlines how the data are handled. Section 4 describes the datasets that are available for download from The Environmental Information Data Centre (EIDC) online data repository. A selection of existing and potential data applications are discussed in Sect. 5, followed by conclusions.

## 2 Measurement methodology

### 2.1 Network creation

Between 2013 and the time of writing, UKCEH has deployed 51 COSMOS-UK environmental monitoring sites across the UK (Fig. 1) (Boorman et al., 2020). Two sites, Wytham Woods and Redmere, have been decommissioned during this time due to changes to site conditions and access. A summary of each site's main characteristics is included in Table 1, and a record of any changes to site land cover is provided in Table 2.


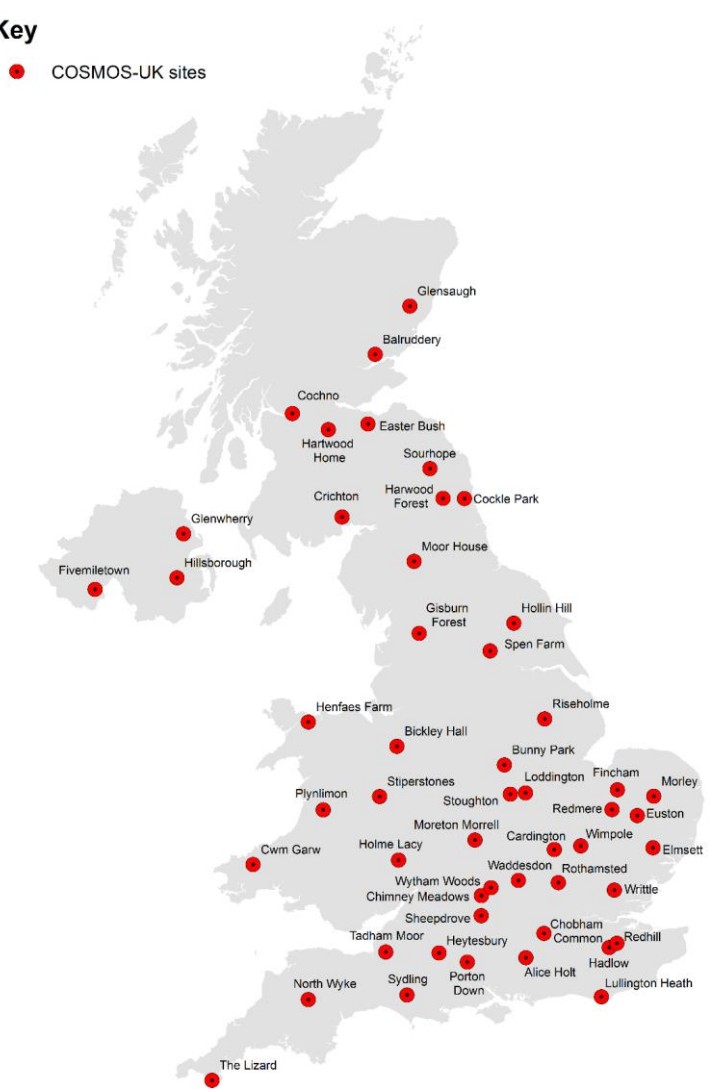

**Figure 1: Map of COSMOS-UK site locations (Boorman et al., 2020).**


**Table 1: Site information. Standard Average Annual Rainfall (SAAR) is provided by the Flood Estimation Handbook (FEH) catchment descriptor SAAR6190 as described in Bayliss (1999). Monitored soil types include: mineral soil (MS), calcareous mineral soil (CMS), organic soil (OS), and organic soil over mineral soil (OSMS). Land covers comprise: broadleaf woodland (BW), coniferous woodland (CW), coniferous forest (CF), arable and horticulture (AH), grassland (G), improved grassland (IG), acid grassland (AG), calcareous grassland (CG), heather grassland (HG), heather (H), and orchard (O).**

| Site name | Start (end) date | Altitude (m) | SAAR (mm) | Soil type | Current land cover |
|---|---|---|---|---|---|
| Alice Holt | 06/03/2015 | 80 | 801 | MS | BW |
| Balruddery | 15/05/2014 | 130 | 740 | MS | AH |
| Bickley Hall | 28/01/2015 | 78 | 727 | MS | G |
| Bunny Park | 27/01/2015 | 39 | 579 | MS | AH |
| Cardington | 24/06/2015 | 29 | 552 | MS | IG |
| Chimney Meadows | 02/10/2013 | 65 | 626 | CMS | IG |
| Chobham Common | 24/02/2015 | 47 | 662 | OSMS | HG |
| Cochno | 23/08/2017 | 168 | 1387 | MS | IG |
| Cockle Park | 21/11/2014 | 87 | 720 | MS | AH |
| Crichton | 02/12/2014 | 42 | 1051 | MS | AH |
| Cwm Garw | 29/06/2016 | 299 | 1740 | MS | IG |
| Easter Bush | 13/08/2014 | 208 | 798 | MS | IG |
| Elmsett | 11/08/2016 | 76 | 564 | CMS | AH |
| Euston | 31/03/2016 | 18 | 600 | MS | IG |
| Fincham | 07/06/2017 | 15 | 613 | CMS | AH |
| Fivemiletown | 26/06/2018 | 174 | 1227 | MS | AH |
| Gisburn Forest | 15/08/2014 | 246 | 1485 | MS | CW |
| Glensaugh | 14/05/2014 | 399 | 1109 | OS | H |

| | | | | | |
|---|---|---|---|---|---|
| Glenwherry | 15/06/2016 | 274 | 1340 | OS | IG |
| Hadlow | 27/10/2016 | 33 | 669 | MS | IG |
| Hartwood Home | 20/05/2014 | 225 | 946 | MS | IG |
| Harwood Forest | 22/05/2015 | 300 | 895 | OS | CF |
| Henfaes Farm | 17/12/2015 | 287 | 1282 | MS | AG |
| Heytesbury | 16/08/2017 | 166 | 832 | CMS | CG |
| Hillsborough | 14/06/2016 | 146 | 909 | MS | IG |
| Hollin Hill | 25/03/2014 | 82 | 673 | MS | IG |
| Holme Lacy | 11/04/2018 | 76 | 674 | MS | AH |
| Loddington | 26/04/2016 | 186 | 664 | MS | AH |
| Lullington Heath | 16/12/2014 | 119 | 825 | CMS | CG |
| Moor House | 04/12/2014 | 565 | 1239 | MS | AG |
| Moreton Morrell | 15/11/2018 | 53 | 611 | MS | IG |
| Morley | 14/05/2014 | 55 | 620 | MS | AH |
| North Wyke | 16/10/2014 | 181 | 979 | MS | AH |
| Plynlimon | 05/11/2014 | 542 | 2421 | OS | AG |
| Porton Down | 18/12/2014 | 146 | 759 | CMS | IG |
| Redhill | 18/02/2016 | 91 | 656 | CMS | O |
| Redmere | 10/02/2015 (-20/09/2018) | 3 | 559 | OS | AH |
| Riseholme | 04/05/2016 | 53 | 603 | CMS | IG |
| Rothamsted | 25/07/2014 | 131 | 692 | MS | AH |
| Sheepdrove | 24/10/2013 | 170 | 737 | MS | AH |

| Sourhope | 19/11/2014 | 487 | 1009 | MS | AG |
| Spen Farm | 23/11/2016 | 57 | 654 | CMS | AH |
| Stiperstones | 06/11/2014 | 432 | 874 | OS | IG |
| Stoughton | 19/08/2015 | 130 | 641 | MS | AH |
| Sydling | 27/11/2018 | 249 | 1064 | MS | IG |
| Tadham Moor | 14/10/2014 | 7 | 749 | OS | IG |
| The Lizard | 17/10/2014 | 85 | 1084 | MS | G |
| Waddesdon | 04/11/2013 | 98 | 636 | MS | IG |
| Wimpole | 10/09/2019 | 30 | 555 | MS | AH |
| Writtle | 04/07/2017 | 44 | 571 | MS | IG |
| Wytham Woods | 26/11/2013 (-01/10/2016) | 109 | 647 | MS | BW |


**Table 2: Changes in land cover at COSMOS-UK sites.**

| Site ID | Land Cover | Land Cover Start Date |
| --- | --- | --- |
| Crichton | IG | 21/11/2014 |
| | AH | 10/05/2019 |
| North Wyke | IG | 16/10/2014 |
| | AH | 09/09/2019 |
| Sheepdrove | IG | 24/10/2013 |
| | AH | 03/10/2019 |

The selection of sites within the network has aimed to provide an appropriate spatial coverage for improving understanding of UK soil moisture conditions, including representation of key land cover and soil types. All UK regions are represented, though there are more sites in the south and east of the UK to adequately capture the greater soil moisture variability in these areas. Installation of sites in less represented regions is in consideration but is dependent on the availability of resources.


Specific site locations have been further determined by practical considerations such as long-term permission and reasonable access for instrument installation and maintenance, and mobile phone network coverage. Where possible, site selection has aimed to exploit opportunities for COSMOS-UK data to support independent, existing research projects, e.g. data assimilation

for forecasting and prediction; validation of remote sensing data; and support of other monitoring programmes and activities. Similarly, site selection has aimed to create partnerships with farmers and support agricultural research.

Some site characteristics can limit their suitability for CRNS soil moisture measurement, such as proximity to open water or shallow or perched groundwater (such features should not be present within the CRNS measurement footprint), and highly variable topography. Sites have therefore been installed in non-mountainous and largely flat locations with no regular irrigation

or close proximity to significant water bodies.

## 2.2 Site data acquisition

Instrumentation at COSMOS-UK sites is largely standardised (Fig. 2), however differences have arisen for the following reasons.

- When instrument performance was reviewed resulting in subsequent installations utilising different, higher-

performance sensors (e.g. for improved sensor accuracy).

- Where a site has been located in an area which is expected to experience a significant period of snow cover, the monitoring equipment includes additional sensors for measurements of snow.

- Where a site has been located within a forest and requires measurements from a tower structure above the canopy of mature vegetation.

These site differences are detailed in Table 3. For further information regarding individual instruments, a detailed summary is provided in the COSMOS-UK User Guide (Boorman et al., 2020).

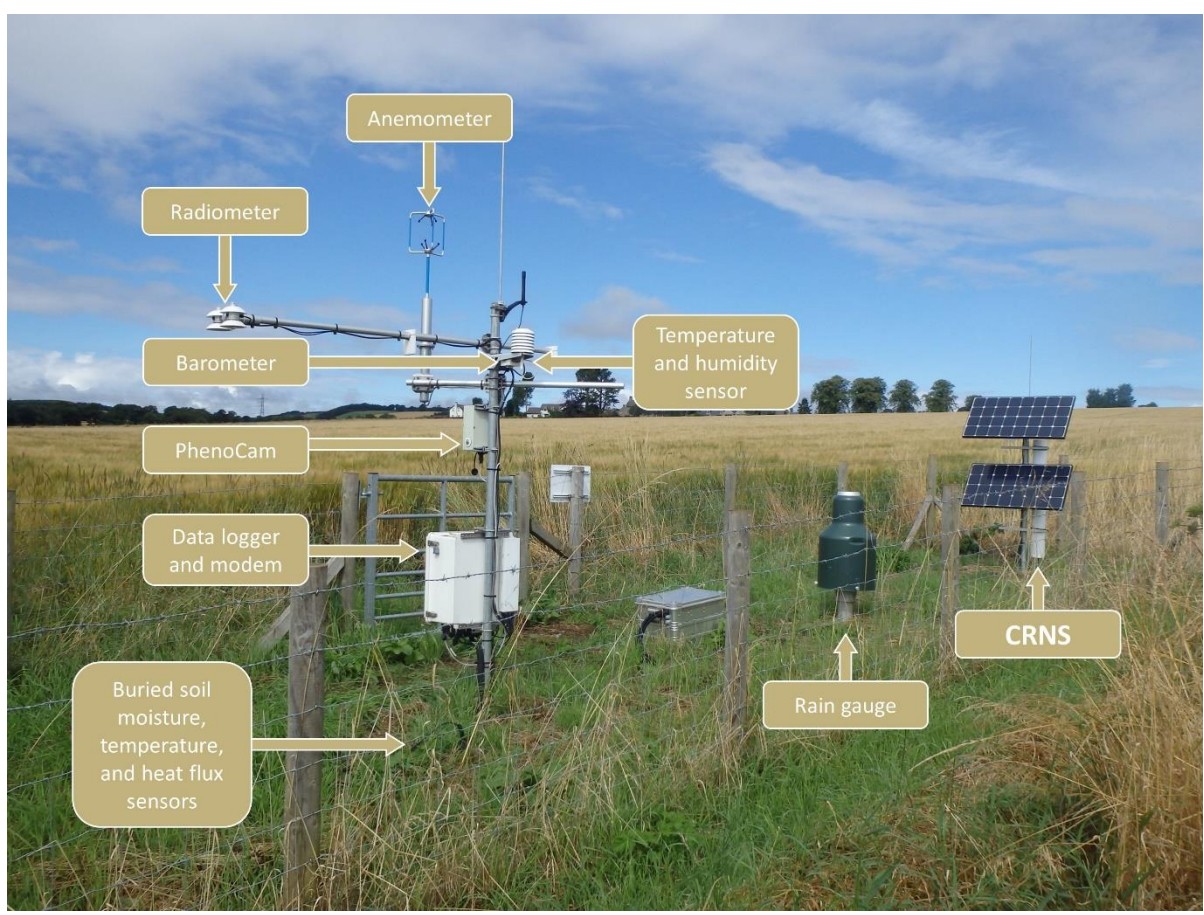

**Figure 2: COSMOS-UK site instrument layout. Photograph of the Balruddery site in Scotland. Photograph taken by Jenna Thornton.**


**Table 3: COSMOS-UK site instrumentation. The snow depth SR50A sensors and SnowFox CRNS are only at sites established as sites experiencing significant periods of snow cover.**

| Data | Instrument | | |
|---|---|---|---|
| Neutron counts and field scale soil moisture | Hydroinnova CRS1000/B CRNS | OR | Hydroinnova CRS2000/B CRNS |
| Point soil moisture and temperature | 2 Acclima ACC-SEN-SDI (TDT) | | |
| Profile soil moisture and temperature | 1 IMKO PICO-PROFILE and 1 Hukseflux STP01 | OR | 8 Acclima ACC-SEN-SDI (TDT) and 1 Hukseflux STP01 |
| Point soil heat flux | 2 Hukseflux HFP01SC | | |

| Precipitation | 1 OTT Pluvio$^2$(L) | OR | 1 OTT Pluvio$^2$(L) and 1 SBS500 |
|---|---|---|---|
| Short- and long-wave radiation in and out | Hukseflux NR01 | | |
| Air temperature and relative humidity | Rotronic HC2(A-)S3 | OR | Vaisala HMP155(A) |
| Barometric pressure | Gill MetPak Pro Base Station | OR | Vaisala PTB110 |
| Wind speed and direction | Gill Integrated WindSonic | OR | Gill WindMaster 3D Sonic Anemometer |
| PhenoCam photos | Mobotix S14 OR S15 OR S16 IP camera | | |
| Snow depth | Campbell Scientific SR50A | | |
| Neutron counts for snow water equivalent | Hydroinnova SnowFox CRNS | | |

Available measurements are described below, and further information regarding variables and recording intervals is provided in Sect. 4. All COSMOS-UK measurements are logged on a CR3000 Micrologger (Campbell Scientific Ltd., Logan, Utah, USA) and telemetered via the 2G, 3G or 4G mobile network, or Inmarsat BGAN satellite network (Inmarsat Global Ltd., London, UK), to secure servers at UKCEH Wallingford. Telemetry has been achieved using either a COM110 (Campbell Scientific Ltd., Logan, Utah, USA), Maestro M100 (Lantronix Inc., Irvine, California, USA), Proroute® H820 (E-Lins Group, Shenzhen, China), or 9502 BGAN (Hughes Network Systems LLC, Germantown, Maryland, USA) modem.

Sensor calibration coefficients are stored on the CR3000 for measurements such as soil heat flux (G, W m$^{-2}$) and the four components of net radiation (RN, W m$^{-2}$). Equipment across the network is promptly replaced when faults are detected, and instruments are tested and re-calibrated on an annual basis under a maintenance contract with the suppliers of the field instrumentation, Campbell Scientific Ltd. A full record of sensor exchanges is maintained by UKCEH.

### 2.2.1 Soil data

Each COSMOS-UK site utilises a moderated CRS2000/B CRNS (Hydroinnova LLC, Albuquerque, New Mexico, USA) which counts epithermal neutrons at the land surface. The sites at Chimney Meadows, Sheepdrove and Wytham Woods were installed with a bare and moderated CRS1000/B, and Waddesdon was installed with only a moderated CRS1000/B (Hydroinnova LLC, Albuquerque, New Mexico, USA) (Zreda et al., 2012). All bare CRNSs have subsequently been removed. Wytham Woods was decommissioned in 2016, and in February 2020 CRS2000/B sensors were installed adjacent to the remaining CRS1000/B instruments. The neutron counts from these sensors are used to derive average field scale volumetric water content (VWC, %) of the near-surface soil layer (see Sect. 3.1 for details). Each site includes point scale soil moisture sensors, which estimate

VWC via Time Domain Transmissometry (TDT). These TDT sensors estimate point scale soil moisture by measuring the time taken for an electromagnetic wave to travel along the sensor's closed circuit; this transmission decreases in speed with soil permittivity (Blonquist et al., 2005). Each site includes either two (deployment prior to March 2016) or ten buried ACC-SEN-

SDI TDTs (Acclima Inc., Idaho, USA) to measure small-area soil VWC (%) at defined depths (listed in Table 4). Sites installed prior to March 2016 included a PICO-PROFILE soil moisture sensor (IMKO Micromodultechnik GmbH, Ettlingen, Germany) to measure VWC (%) at depths of 0.15, 0.4 and 0.65 m. The configuration of site sensors resulted in occasional data loss and the PICO-PROFILE instruments were subsequently removed from sites during 2019-2020 network maintenance to improve overall data capture. Soil heat flux (W m$^{-2}$) is measured at every site using a pair of HFP01-SC sensors (Hukseflux Thermal

Sensors B.V., Delft, The Netherlands) buried at a depth of 0.03 m. All sites include an STP01 profile soil temperature sensor (Hukseflux Thermal Sensors B.V., Delft, The Netherlands) to measure the soil temperature gradient (°C) at 0.02, 0.05, 0.1, 0.2 and 0.5 m depths.

**Table 4: Buried depths of the paired TDT point soil sensors. TDT3-10 are only present at sites installed on or after 31 March 2016.**
**At the Heytesbury site TDT9 and 10 are buried at 0.05 m depth due to the presence of solid chalk. TDT pair 1 and 2 are located 1 m apart, whilst the additional TDTs (3-10) are buried with 0.3 m space between the paired sensor and 0.15 m horizontal distance between pairs. Data for each individual sensor are provided.**

| TDT1 | TDT2 | TDT3 | TDT4 | TDT5 | TDT6 | TDT7 | TDT8 | TDT9 | TDT10 |
|------|------|------|------|------|------|------|------|------|-------|
| 0.1 m | 0.1 m | 0.05 m | 0.05 m | 0.15 m | 0.15 m | 0.25 m | 0.25 m | 0.5 m | 0.5 m |

### 2.2.2 Hydrometeorological data

COSMOS-UK sites include a Pluvio$^2$(L) digital weighing rain gauge (OTT HydroMet, Kempten, Germany) installed with an aperture height of 1 m above the soil surface. These rain gauges measure precipitation intensity and amount (mm) at 1-minute resolution. Sites were identified as being not particularly exposed and therefore Pluvio wind shields were not installed. Incoming and outgoing short- and long-wave radiation (W m$^{-2}$) are measured at each site using an NR01 four-component net radiometer (Hukseflux Thermal Sensors B.V., Delft, The Netherlands). Barometric pressure (hPa) is measured at all sites using

either a Gill MetPak Pro Base Station (Gill Instruments Ltd., Lymington, UK) at a height of 2 m or a PTB110 barometer (Vaisala Corporation, Helsinki, Finland). From this, pressure corrected to sea level is derived. Air temperature (°C) and relative humidity (%) are measured at every site using either an HC2(A-)S3 (Rotronic, Bassersdorf, Switzerland) or HMP155(A) sensor (Vaisala Corporation, Helsinki, Finland). Air temperature and relative humidity are measured at the standard height of 2 m. Wind speed and direction are measured using either a 2-dimensional WindSonic at a measurement height of 2.2 m or 3-

dimensional WindMaster anemometer (Gill Instruments Limited, Lymington, UK) at a measurement height of 2.6 m.

### 2.2.3 Non-standard sites

COSMOS-UK sites located in dense forest or woodland (Alice Holt, Harwood Forest and Wytham Woods) were designed with certain meteorological sensors installed above the canopy, on pre-existing flux monitoring towers. Wind measurements, barometric pressure, relative humidity, air temperature, precipitation, and the components of net radiation are measured above the canopy. The measurement height of these variables ranges from approximately 23–33 m. Precipitation is captured by a funnel above the canopy and fed via a tube to the Pluvio$^2$(L) rain gauge located at ground level. Forest sites do not accurately measure rainfall intensity due to the lag time in precipitation captured above canopy and recorded in the rain gauge below. Precipitation data are corrected for the smaller aperture area of the funnel relative to that of the Pluvio$^2$(L).

Across the COSMOS-UK network, eight site locations were identified in areas likely to experience a significant period of snow cover over the winter period. These sites (Glensaugh, Easter Bush, Gisburn Forest, Plynlimon, Sourhope, Moor House, Cwm Garw and Cochno) were installed with two additional sensors: an SR50A snow depth sensor (Campbell Scientific Ltd., Logan, Utah, USA) measuring small area snow depth (mm); and a buried SnowFox CRNS (Hydroinnova LLC, Albuquerque, New Mexico, USA) measuring neutron counts which can be used to derive snow water equivalent (Desilets, 2017).

Tadham Moor is located on the Somerset Levels, an area that can experience inundation during high rainfall. The COSMOS-UK site was therefore adapted to withstand any significant floodwater. For this reason, the digital weighing rain gauge has an aperture height of approximately 1.7 m, and the CRNS is installed horizontally at a height of approximately 1.1 m rather than vertically. This non-standard installation enables an assessment of the CRNS technology in a very high soil moisture environment.

During COSMOS-UK network maintenance in February 2020 an SBS500 tipping bucket rain gauge (Environmental Measurements Limited, North Shields, UK) was added to three sites (Chimney Meadows, Sheepdrove and Waddesdon), providing an additional precipitation (mm) reference against which the performance of the Pluvio$^2$(L) rain gauges can be evaluated. The SBS500 tipping bucket rain gauge (TBR) was chosen for its improved aerodynamic characteristics and reduction in turbulence and under-catch (Colli et al., 2018; Strangeways, 2004).

### 2.3 Soil sampling and lab analysis for site calibration

An in situ soil sampling procedure adapted from Franz (2012) and Zreda et al. (2012) has been completed at each COSMOS-UK site following installation. The results from the sampling are used to determine site-specific soil properties for CRNS calibration: field average soil moisture and dry bulk density, lattice and bound water, and organic matter. Once the CRNS count data have been corrected for atmospheric pressure (Desilets, 2017; Evans et al., 2016), humidity (Evans et al., 2016; Rosolem et al., 2013) and an empirical background neutron intensity factor (adapted from Evans et al., 2016), the calibration data are used to derive $N_0$ on the day of calibration (details in Sect. 3.1). Soil samples for determination of VWC and dry bulk density were taken at 18 representative locations centred on the CRNS: at compass bearings of 0, 60, 120, 180, 240 and 300 degrees and at 5, 25 and 75 m radial distance at each of these compass bearings (Fig. 3). For CRNS calibrations

before 14 September 2016, samples were taken at 25, 75 and 200 m radial distances. These locations follow Franz (2012), subsequently modified to account for revised CRNS footprint characteristics (Köhli et al., 2015b). In addition, as the 180

210  degree sample at 5 m distance would fall on a cable run within the CRNS enclosure, this location has been replaced with a sample at either 90 or 270 degrees at 1 m distance. At each location volumetric soil samples (using 0.05 m diameter, 0.051 m length rings (Eijkelkamp 07.53.SC sample ring kit and Edelman auger)) were taken at five depths: 0–0.05, 0.05–0.1, 0.1–0.15, 0.15–0.2 and 0.2–0.25 m below ground level (bgl). Soil sampling depths for CRNS calibration were selected to match typical (moist) UK conditions, and higher weighting is later applied to shallow soil layers to ensure appropriate representation of

215  decreasing contribution of deeper water (Köhli et al., 2015a; Schrön et al., 2017) (see Sect. 3.1 for details). Three locations at different bearings and distances were also selected for an additional soil sample for the determination of lattice and bound water and organic matter. The additional soil samples were taken from 0–0.25 m bgl. This therefore gives a total of 90(+3) soil samples for each calibration.

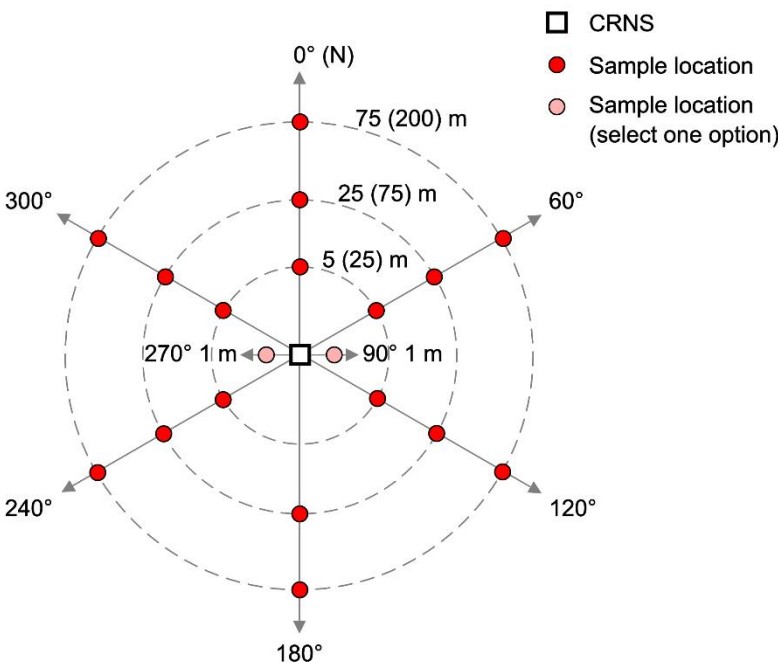

220  **Figure 3: Plan view of soil sampling locations (not to scale). Distances used prior to 14-09-2016 are shown in brackets.**

The field soil samples were returned to the laboratory for analysis. VWC and dry bulk density were determined for the 90 volumetric samples using oven drying (~36 hours at 105 °C). Following analysis, a ~2 g sub-sample was taken from each sample and aggregated to form a composite sample for lattice and bound water and organic matter determination. The three

additional soil samples from the field were air dried (on the lab bench or in the oven at 30 °C) for around three days. The additional samples, along with the composite, were then crushed to pass a ~0.4 mm sieve and subsequently air dried at 105 °C for ~36 hours. Soil organic matter was then estimated for a ~3 g air dried sub-sample (with 6 replicates per additional sample, i.e. 24 sub-samples) using loss on ignition at 400 °C for 16 hours in the furnace (following Nelson and Sommers, 1996). Following cooling in a desiccator and weighing, the sub-samples were then returned to the furnace to estimate lattice and bound water by loss on ignition at 1000 °C for 4 hours (following Pansu and Gautheyrou, 2006). For use in the CRNS calibration calculation, soil organic carbon was estimated as 50 % of soil organic matter (Nelson and Sommers, 1996). Pansu and Gautheyrou (2006) note that loss on ignition removes organic matter at 300–500 °C and lattice and bound water at 350–1000 °C. The procedure outlined above therefore follows the 400 °C temperature recommendation by Nelson and Sommers (1996), which removes organic matter but causes minimal dehydroxylation of clay minerals. The CRNS calibration procedure uses the mean soil organic carbon and mean lattice and bound water from the 24 sub-samples along with the mean dry bulk density from the 90 volumetric samples. The field average reference VWC for the day of calibration is then calculated as a radial and vertical weighted mean following Köhli et al. (2015). Planned work includes obtaining site bulk density using this weighting function. The soil properties and soil moisture results for calibrating each site are available in Table 5.

Repeat calibrations using secondary samples have been conducted at two COSMOS-UK sites to explore the accuracy of the derived VWC obtained on a particular day using this methodology. There was < 0.03 cm$^3$ cm$^{-3}$ difference in VWC between the soil moisture determined from the second calibration and the corresponding daily VWC value derived using the site's first calibration. Considering the estimated errors in soil sampling and (to a lesser extent) laboratory procedures, the difference in calibrations is considered to be within the uncertainty of the reference soil moisture determined from secondary sampling and the predicted VWC from the CRNS and its original calibration. Additional repeat calibrations are planned across the network to help further analyse the current methodologies and assess sensor performance over time.

Table 5: COSMOS-UK soil sampling results. Standard deviations are available in the dataset (Stanley et al., 2021).

| Site name | Date of calibration | Reference soil moisture (cm$^3$ cm$^{-3}$) | Reference bulk density (g cm$^{-3}$) | Reference lattice water (g g$^{-1}$) | Reference soil organic carbon (g g$^{-1}$) |
|---|---|---|---|---|---|
| Alice Holt | 04/08/2015 | 0.266 | 0.85 | 0.025 | 0.042 |
| Balruddery | 29/07/2014 | 0.254 | 1.34 | 0.018 | 0.023 |
| Bickley Hall | 24/02/2015 | 0.412 | 1.31 | 0.010 | 0.020 |
| Bunny Park | 25/02/2015 | 0.283 | 1.55 | 0.008 | 0.016 |
| Cardington | 18/08/2015 | 0.141 | 1.14 | 0.016 | 0.040 |
| Cardington | 17/01/2018 | 0.325 | 1.30 | 0.014 | 0.032 |
| Chimney Meadows | 13/11/2013 | 0.393 | 1.36 | 0.011 | 0.027 |

| | | | | | |
|---|---|---|---|---|---|
| Chimney Meadows | 31/08/2018 | 0.247 | 1.26 | 0.011 | 0.032 |
| Chobham Common | 12/03/2015 | 0.566 | 0.90 | 0.003 | 0.031 |
| Cochno | 18/10/2017 | 0.524 | 0.83 | 0.019 | 0.068 |
| Cockle Park | 10/12/2014 | 0.447 | 1.21 | 0.020 | 0.033 |
| Crichton | 08/12/2014 | 0.428 | 1.15 | 0.011 | 0.045 |
| Cwm Garw | 28/09/2018 | 0.417 | 0.96 | 0.022 | 0.048 |
| Easter Bush | 16/09/2014 | 0.303 | 1.10 | 0.019 | 0.033 |
| Elmsett | 19/01/2017 | 0.400 | 1.26 | 0.015 | 0.022 |
| Euston | 18/01/2017 | 0.189 | 1.27 | 0.003 | 0.029 |
| Fincham | 28/07/2017 | 0.279 | 1.33 | 0.007 | 0.02 |
| Fivemiletown | 15/11/2018 | 0.537 | 0.97 | 0.014 | 0.039 |
| Gisburn Forest | 17/09/2014 | 0.542 | 0.82 | 0.021 | 0.061 |
| Glensaugh | 28/07/2014 | 0.608 | 0.44 | 0.014 | 0.203 |
| Glenwherry | 20/10/2016 | 0.631 | 0.54 | 0.024 | 0.153 |
| Hadlow | 15/12/2016 | 0.398 | 1.22 | 0.028 | 0.031 |
| Hartwood Home | 30/07/2014 | 0.356 | 1.02 | 0.033 | 0.043 |
| Harwood Forest | 14/06/2017 | 0.591 | 0.33 | 0.009 | 0.304 |
| Henfaes Farm | 06/10/2016 | 0.507 | 0.97 | 0.022 | 0.077 |
| Heytesbury | 22/02/2018 | 0.411 | 0.88 | 0.006 | 0.066 |
| Hillsborough | 19/10/2016 | 0.450 | 1.15 | 0.021 | 0.042 |
| Hollin Hill | 25/06/2014 | 0.364 | 1.06 | 0.025 | 0.032 |
| Holme Lacy | 03/05/2018 | 0.292 | 1.24 | 0.017 | 0.022 |
| Loddington | 14/09/2016 | 0.455 | 1.16 | 0.041 | 0.036 |
| Lullington Heath | 14/01/2015 | 0.452 | 0.90 | 0.006 | 0.043 |
| Moor House | 11/12/2014 | 0.578 | 0.76 | 0.014 | 0.076 |
| Moreton Morrell | 13/02/2019 | 0.433 | 1.22 | 0.026 | 0.035 |
| Morley | 19/06/2014 | 0.161 | 1.53 | 0.016 | 0.017 |
| North Wyke | 05/11/2014 | 0.472 | 1.12 | 0.02 | 0.037 |
| Plynlimon | 26/11/2014 | 0.590 | 0.62 | 0.02 | 0.098 |
| Porton Down | 02/02/2015 | 0.391 | 0.97 | 0.004 | 0.049 |
| Redhill | 08/12/2016 | 0.252 | 1.26 | 0.011 | 0.024 |
| Redmere | 04/06/2015 | 0.504 | 0.60 | 0.056 | 0.238 |
| Riseholme | 16/02/2017 | 0.429 | 1.27 | 0.022 | 0.032 |

| | | | | | |
|---|---|---|---|---|---|
| Rothamsted | 02/09/2014 | 0.280 | 1.33 | 0.018 | 0.021 |
| Sheepdrove | 20/03/2014 | 0.327 | 1.04 | 0.027 | 0.059 |
| Sourhope | 09/12/2014 | 0.578 | 0.65 | 0.021 | 0.086 |
| Spen Farm | 15/06/2017 | 0.269 | 1.41 | 0.011 | 0.019 |
| Stiperstones | 27/11/2014 | 0.612 | 0.62 | 0.016 | 0.104 |
| Stoughton | 19/11/2015 | 0.351 | 1.33 | 0.018 | 0.027 |
| Sydling | 21/03/2019 | 0.374 | 1.17 | 0.020 | 0.035 |
| Tadham Moor | 06/11/2014 | 0.615 | 0.32 | 0.029 | 0.314 |
| The Lizard | 04/11/2014 | 0.568 | 0.95 | 0.014 | 0.058 |
| Waddesdon | 13/03/2014 | 0.460 | 1.11 | 0.021 | 0.034 |
| Wimpole | 15/10/2019 | 0.361 | 1.22 | 0.015 | 0.035 |
| Writtle | 27/07/2017 | 0.350 | 1.26 | 0.019 | 0.035 |
| Wytham Woods | 15/04/2014 | 0.485 | 1.05 | 0.017 | 0.028 |

## 3 COSMOS-UK data

### 3.1 Deriving soil moisture from the CRNS

Field scale soil moisture (CRNS VWC) is derived from the corrected CRNS epithermal neutron counts, which inversely correlate with hydrogen present at the land surface (soil, vegetation and any other water sources) (Zreda et al., 2008; Zreda et al., 2012). Incoming epithermal neutrons collide with hydrogen nuclei at the land surface and are therefore moderated by the hydrogen present in water molecules, thereby enabling an indirect measurement of soil moisture (Rivera Villarreyes et al., 2011). VWC is calculated using the following formula where coefficients are determined for a basic silica soil (Desilets et al., 2010; Evans et al., 2016),

$$VWC = \left( \frac{0.0808}{\left( \frac{N_{corr}}{N_0} \right) - 0.372} - 0.115 - (\tau + SOC) \right) \frac{\rho_{bd}}{\rho_w} \tag{1}$$

In Eq. (1), $N_{corr}$ are the corrected counts, $\tau$ is the reference lattice and bound water, SOC is the reference soil organic carbon, $\rho_{bd}$ is the reference bulk density and $\rho_w$ is the water density equal to 1 g cm$^{-3}$. $\tau$, SOC and $\rho_{bd}$ are determined on the calibration day by field and laboratory analysis (Evans et al., 2016; Franz, 2012; Franz et al., 2013; Zreda et al., 2012). $N_{corr}$ is obtained by aggregating raw neutron counts from each site to a 60-minute interval and correcting for atmospheric pressure (Desilets, 2017; Evans et al., 2016), humidity (Evans et al., 2016; Rosolem et al., 2013) and background neutron intensity variations (adapted from Evans et al., 2016) using in situ measurements. The atmospheric pressure correction uses instantaneous

barometric attenuation lengths (Desilets and Zreda, 2003) calculated for COSMOS-UK sites according to crnslab.org/util/intensity.php and the correction uses a fixed reference pressure value of 1000 hPa. The background neutron intensity correction uses the publically available Jungfraujoch (JUNG) data (nmdb.eu/station/jung/) provided by the Physikalisches Institut, University of Bern, Switzerland. Normalised count rates from JUNG are retrieved and used in sub-daily calculations to produce near-real time COSMOS-UK datasets; the period of record is subsequently updated for any changes to JUNG data on an annual basis. Where data are unavailable from the JUNG detector the period is infilled with appropriately scaled values from alternate monitors: another counter at Jungfraujoch (JUNG1), Newark in the USA (NEWK) provided by the University of Delaware Department of Physics and Astronomy and the Bartol Research Institute, or Apatity in Russia (APTY). When choosing the most suitable neutron monitors for COSMOS-UK data, these monitors were identified as well-maintained with high levels of data completeness. The geomagnetic cut-off rigidity of the available monitors' locations was also considered when identifying suitable monitors. Normalised count rates are not greatly affected by cut-off rigidity except for during significant space weather events, when magnetic field disturbances may result in a change to a location's cut-off rigidity. A comparison between JUNG and monitors with cut-off rigidities similar to COSMOS-UK sites presented good agreement between the normalised counts and associated trends. Following this correction for background neutron intensity, counts are then calibrated to the site's specific soil, using the soil calibration values determined by UKCEH laboratory analysis.

COSMOS-UK uses the site-specific $N_0$ method (Desilets et al., 2010) for deriving water content from a site's corrected neutron count data, where $N_0$ is the site-specific neutron counting rate over dry soil under reference atmospheric pressure and solar activity conditions. Alternative methods are described in Baatz et al. (2014), Bogena et al. (2015) and Iwema et al. (2015). A site-specific $N_0$ value is calculated by rearranging Eq. (1) for $N_0$ and substituting the average neutron counts on the day of calibration for N, together with reference soil moisture for VWC. The corrected counts and $N_0$ can then be input into Eq. (1) to produce VWCs. These data are subsequently constrained to the physical range of 0–100 % soil water content by determining values of $N_{max}$ and $N_{min}$ respectively, the maximum and minimum physically admissible neutron count value for each site. Figure 4 shows an example of the calibration curve for the Redhill site, located in South East England.

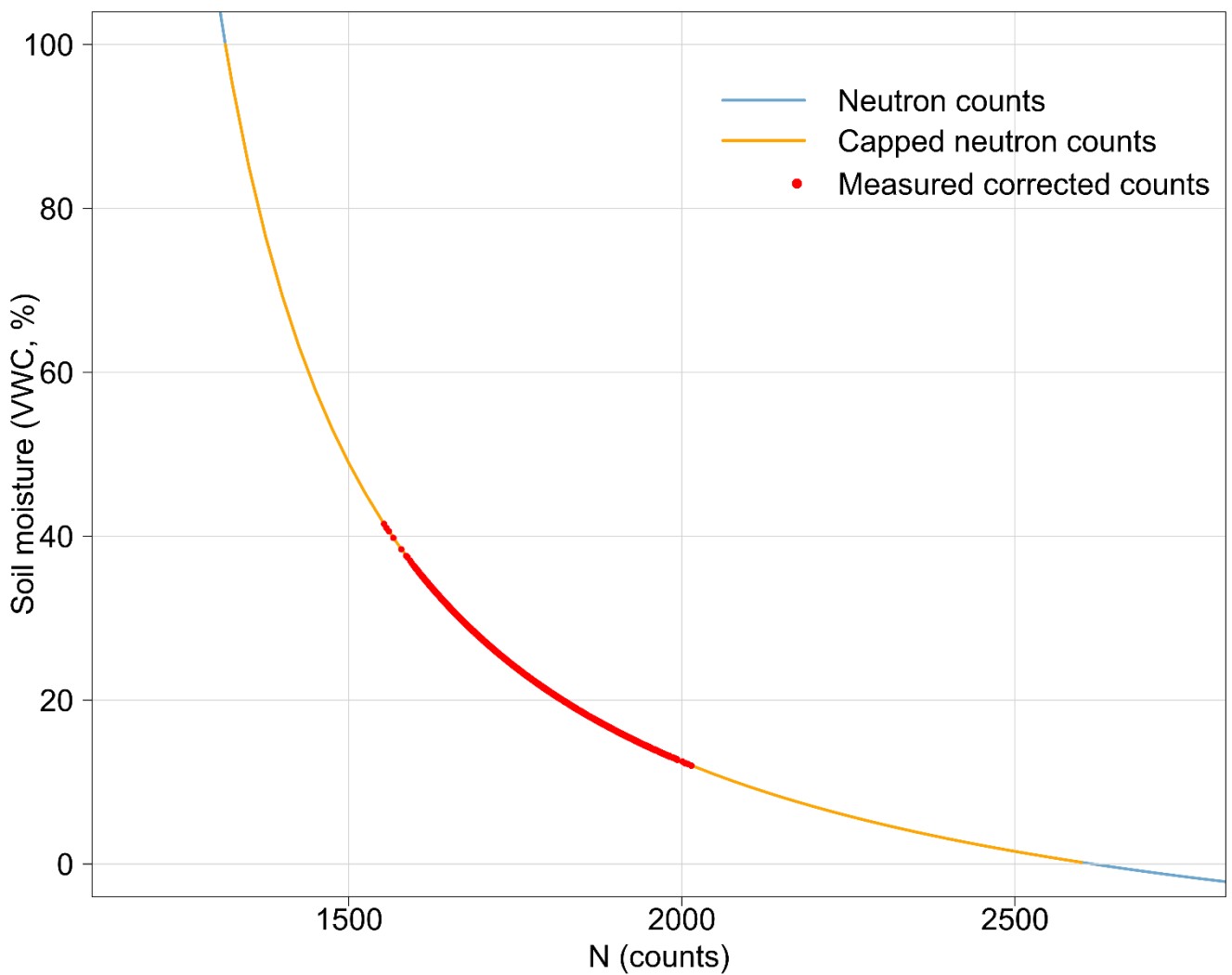

**Figure 4: The calibration curve (Eq. (1)) for determining soil moisture for the Redhill site. The range of neutron counts and the derived CRNS VWC is shown in blue. The range of possible capped neutron counts and VWC (determined by $N_{min}$ and $N_{max}$) is shown in orange. Corrected neutron counts and corresponding VWC at this site between 18 February 2016 and 08 February 2021 is shown in red.**

Once complete, this process produces the hourly CRNS VWC dataset. In a subsequent process, hourly corrected neutron counts are averaged to a daily mean and undergo the same calculations to produce the daily CRNS VWC soil moisture dataset. A minimum of 20 hourly values in a day is set as the requirement to produce a daily soil moisture value. An additional version of the soil moisture dataset is calculated, in which daily CRNS VWC has been adjusted for snow events using site measurements of albedo.

An in-lab cross-comparison was performed on the majority of CRNSs prior to field deployment. Cross-calibration of deployed CRNSs was also carried out at six COSMOS-UK sites; data were captured from two adjacent CRNSs for a period of several months to establish a reliable relationship between their counts using a linear regression model.

Point soil moisture and precipitation data at each COSMOS-UK site provide important ancillary information for assessing the potential accuracy of the CRNS VWC data. Figure 5 shows each of the processing stages for deriving soil water content from

neutron counts for the Cochno site in Scotland, alongside soil moisture measured by the 10 buried point sensors and precipitation. This figure clearly shows that daily CRNS VWC data closely resemble the soil moisture dynamics measured by the point sensors, and the response of both VWC measurements to precipitation events.

Some sites may have a higher CRNS VWC measurement uncertainty. For example sites with extensive soil organic matter accumulation (e.g. carbon-dense peatlands) or mature woodlands where CRNS VWC methods might need to be further refined

to account for biomass, plant roots, litter-layer thickness and intercepted water (Andreasen et al., 2017; Baatz et al., 2015; Heidbüchel et al., 2016; Rivera Villarreyes et al., 2011). The contrast of CRNS VWC measurements between sites can be seen in Fig. 6, which displays all data for the period of record as a normalised curve for each site. This figure demonstrates the importance of identifying and understanding localised soil properties, and shows how sites in close proximity and experiencing broadly similar weather patterns can exhibit vastly different ranges and extremes in VWC.


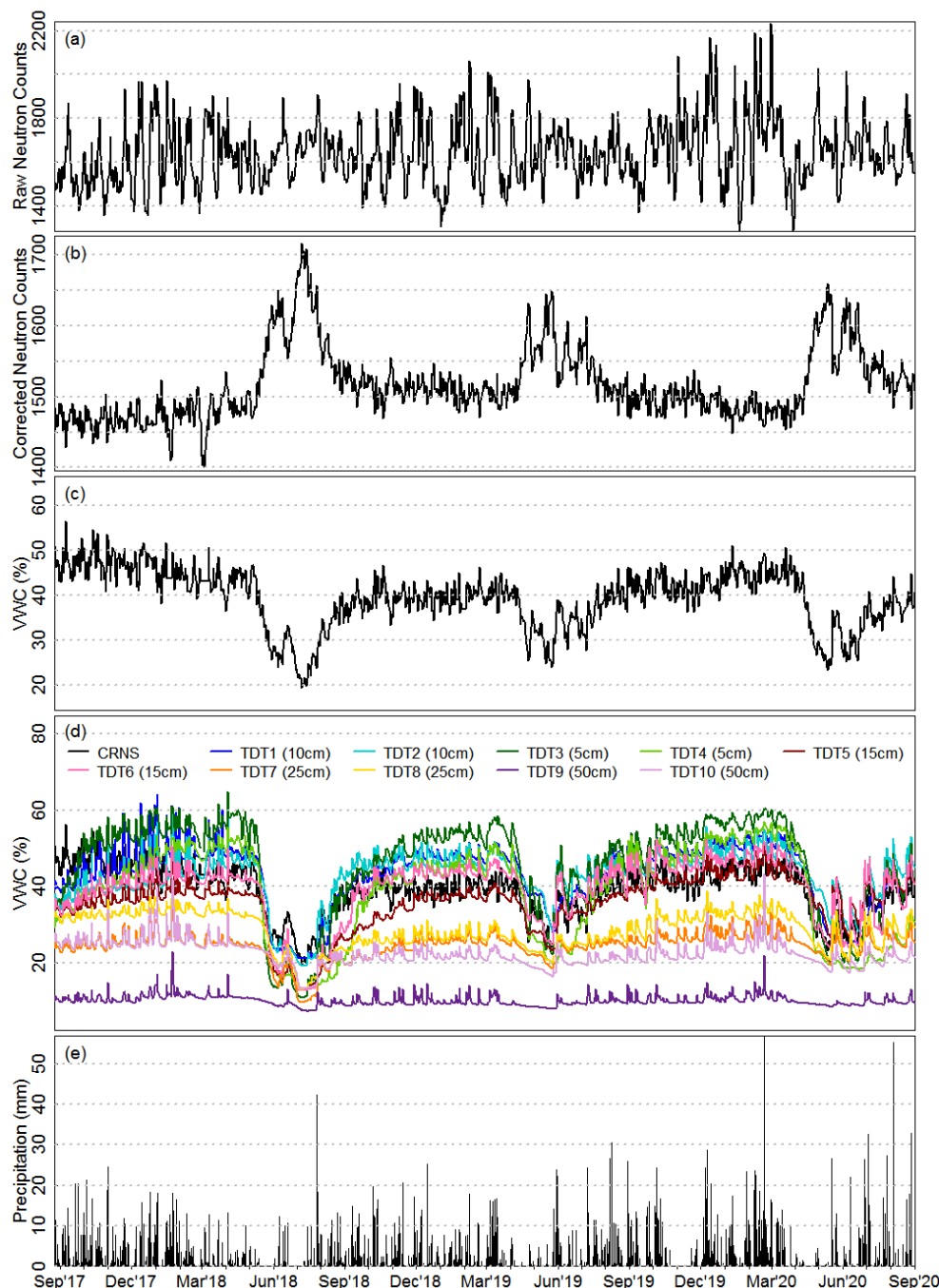

**Figure 5: Daily COSMOS-UK data for the Cochno site in Scotland. (a) Raw neutron counts from the CRNS (aggregated from hourly totals); (b) neutron counts corrected for pressure, humidity and background count intensity (aggregated from hourly totals); (c) VWC determined from the CRNS corrected counts and corrected for snow; (d) CRNS VWC and point TDT VWC at a series of depths; (e) precipitation. Note the 2018 cold wave and summer heatwave impact on soil moisture.**

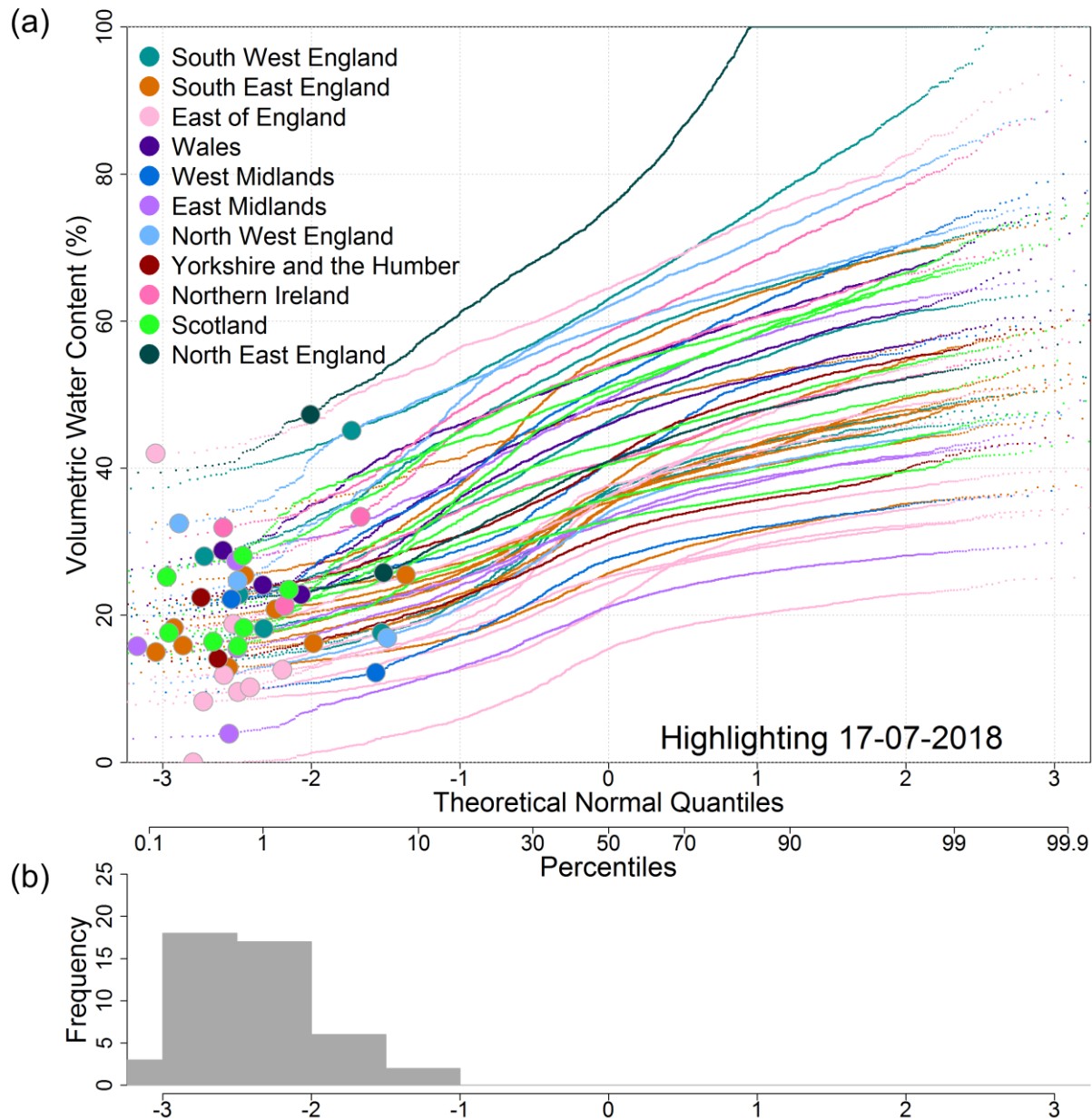

**Figure 6: Soil moisture regime plot for all COSMOS-UK sites grouped by region (dot colour) according to the Nomenclature of Territorial Units for Statistics (NUTS) codes of the United Kingdom. The (a) dots and (b) histogram represent the soil moisture and corresponding frequency, respectively, on 17 July 2018 when there was a widespread drought across the UK. Each line represents the distribution of CRNS VWC at a COSMOS-UK site; sites with wetter regimes plot higher up in the figure.**


**3.2 Soil moisture measurement area and depth**

The CRNS VWC value is an average soil moisture measurement (%) across an estimated, variable footprint of radius up to 200 m, and estimated, variable measurement depth of between approximately 0.1 and 0.8 m (following Köhli et al., 2015). Measurement area depends on local soil moisture, humidity and land cover (Köhli et al., 2015b), whilst penetration depth depends on soil moisture as well as lattice water and soil organic matter water equivalent (Zreda et al., 2008; Franz et al., 2012; Zreda et al., 2012). The greater the actual soil water content, the smaller the CRNS measurement area and shallower the penetrative depth. The measurement area of the CRNS was initially believed to have a radius of approximately 300 m (Zreda et al., 2008); however Köhli et al., (2015) report that 50 % of measured neutrons originated within 50 m of the CRNS, and the footprint radius extended to only 240 m in arid climates. The penetration depth of the measurement is greatest near the CRNS and decreases with distance from the sensor; this varying depth across the footprint is provided as 'D86', the depth at which 86 % of the measured neutron counts are estimated to have originated at a given distance (Zreda et al., 2008; Franz et al., 2013). In the COSMOS-UK dataset, D86 is provided at distances of 1, 5, 25, 75, 150 and 200 m from the CRNS. Figure 7 shows the estimated D86 values for a typically drier site, Euston (average soil moisture approximately 15 %) and a typically wetter site, Riseholme (average soil moisture approximately 33 %) for 2018. During this year the UK experienced a cold wave with significant snow in February to March and a heatwave in June to August. This figure presents how measurement depth increases in drier conditions, decreases with distance from the CRNS, and differs between sites.




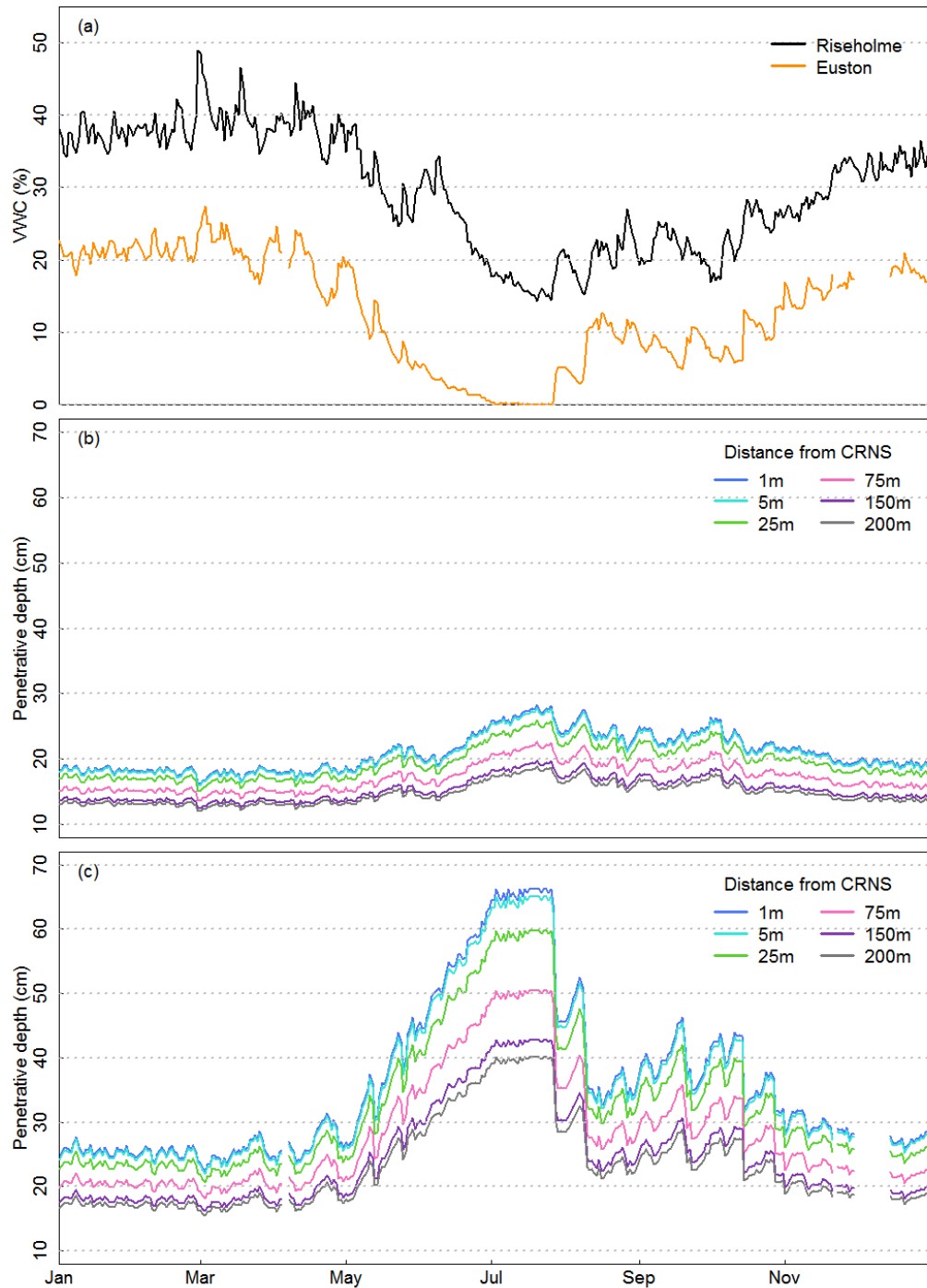


**Figure 7: CRNS VWC and corresponding D86 penetrative depth estimates at a range of distances for two COSMOS-UK sites throughout 2018. a) CRNS VWC for the Riseholme and Euston sites; b) D86 values for Riseholme; c) D86 values for Euston.**

### 3.3 Data processing and quality procedures

Raw data collected at each COSMOS-UK site, comprising the measured variables described above as well as additional diagnostic data from sensors (e.g. internal humidity of the CRNS), are telemetered to UKCEH and stored in an Oracle relational database (Oracle, 2013). When new values are derived following the application of corrections, calibrations and quality tests, these derived data are stored in separate, secondary tables. These secondary datasets are those that are published.

Data quality assurance (QA) and quality control (QC) are applied to specific variables in the raw data. QC is conducted in two

stages:

1. Automated processing is applied to raw data to provide a quality assured dataset. Data which fail the tests are flagged and are not written to secondary datasets. These automated tests include pre-processing for known errors and subsequent QC processes for detecting additional erroneous data. These processes are explained below.

2. Regular manual inspection of raw, diagnostic and processed data is performed using a variety of automated summary

plots and reports. Clearly erroneous data that have passed the automated QC tests are flagged and omitted from the secondary dataset.

Automated processing tasks assess the raw data and create a flagged dataset based on the test results. This enables tracking of data removal and ensures raw data are not lost or overwritten. Raw data are passed through multiple independent QC tests (Table 6). Each test assigns a unique flag value to any raw data which fails. Where data fail multiple tests the flag values are

summed. The summed flag values are unique for each combination of tests, allowing failed tests to be determined from the sum. Where data pass all QC tests, the flag values are assigned '0'. The tests flag issues including: data exceeding known thresholds; implausible values; and data where other variables indicate an issue. The secondary dataset comprises all data not flagged by the QC processes.

All derived datasets are obtained using the quality checked 30 minute data. Planned future work includes the development of

a tertiary dataset comprising quality processed and gap-filled data.

**Table 6: Unique flag values assigned to data based on QC test results.**

| Test | Flag description | Flag value |
|------|------------------|------------|
| | Passes all tests | 0 |
| Missing | Fails the test for missing values | 1 |
| Zero data | Fails the test for zero values where impossible | 2 |
| Too few samples | Fails if not enough samples taken by the data logger over the data interval | 4 |
| Low power | Fails if the site battery level is too low | 8 |
| Sensor faults | Fails where sensor has been manually recorded as faulty for a period of the record | 16 |

| Diagnostic | Fails based on diagnostic data for particular sensors | 32 |
|---|---|---|
| Range | Fails if values are outside a predefined range for the variable | 64 |
| Secondary variables | Fails if a value of one variable implies a fault with another | 128 |
| Spike | Fails where a spike in the data exceeds a given threshold | 512 |
| Error codes | Fails where data contains any known error code | 1024 |

## 3.4 Derived data

In addition to the COSMOS-UK observed soil and hydrometeorological data, the network provides derived datasets including potential evaporation (PE), albedo, snow days, and snow water equivalent (SWE).

PE has been derived from each site's solar radiation, soil heat flux, air temperature, humidity, and wind speed data using the Penman-Monteith method as described by the Food and Agriculture Organization of the United Nations (FAO) (Allen et al., 1998) (Fig. 8). Daily PE data for all COSMOS-UK sites are provided in this dataset.

Snow days have been identified using albedo measurements and SWE has been determined using the albedo and neutron count data available from the CRNS at each COSMOS-UK site. Neutron counts from both the CRNS and SnowFox sensor are sensitive to all sources of water in the environment, allowing them to be used to estimate the SWE held in a snow pack. First the albedo is used to determine the presence or absence of snow cover and then, if present, the reduction in neutron count rate from an estimated snow-free value is used to approximate the SWE, following the method of Desilets (2017). This dataset

includes CRNS SWE. Methods for estimating SWE are available from Wallbank et al. (2020b) and discussed in more detail in Wallbank et al. (2020a).

Available derived data are listed in Sect. 4.

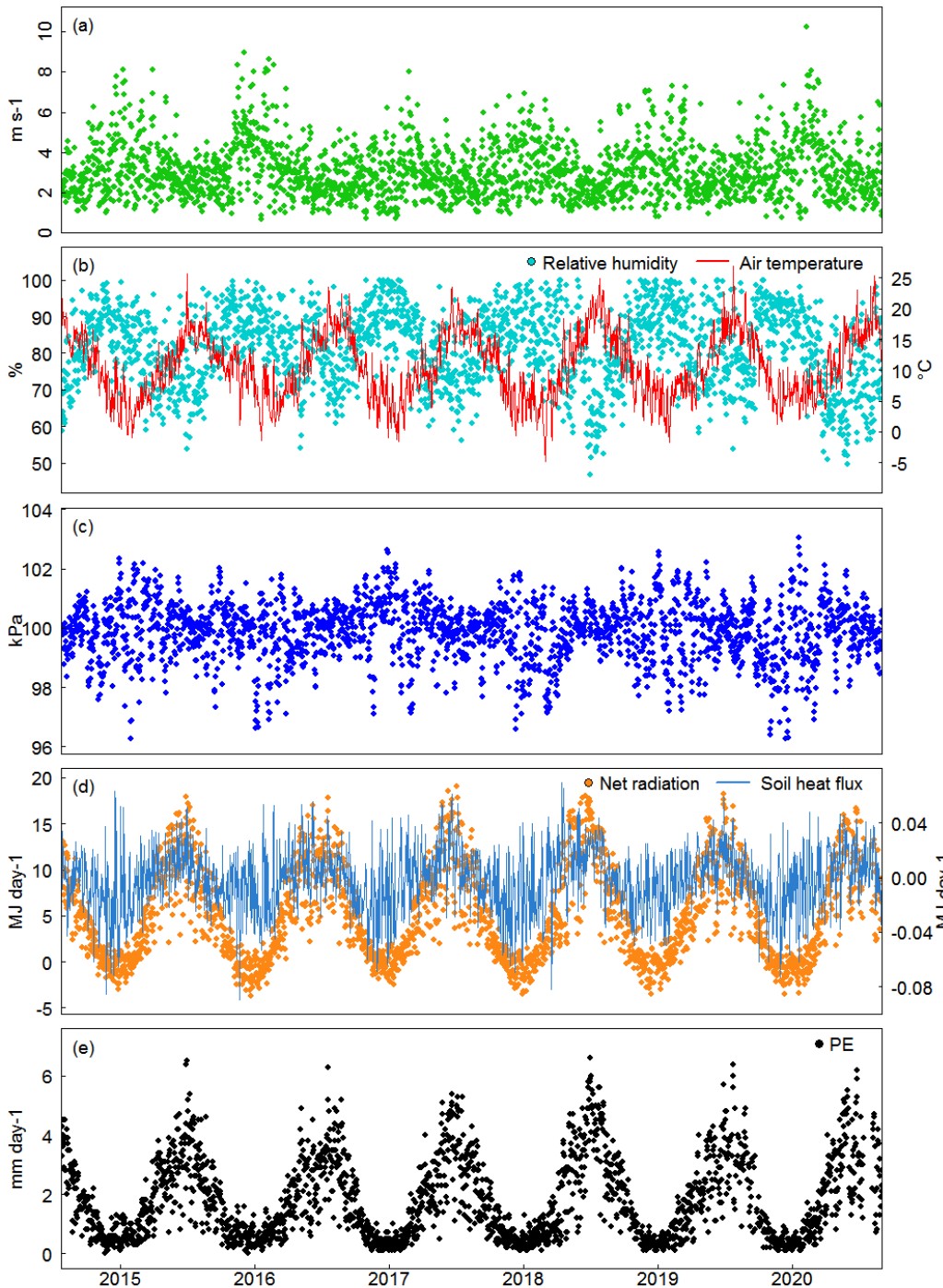

 **Figure 8: Daily COSMOS-UK observations required for the calculation of Potential Evaporation (PE) and derived PE at the Rothamsted site in East Anglia. (a) Wind speed; (b) relative humidity and air temperature; (c) barometric pressure; (d) net radiation and soil heat flux; (e) potential evaporation.**

### 3.5 Additional available data

Additional information can be derived from the data provided by COSMOS-UK sites. As part of ongoing and planned evolution of the network, the additional data described in this section are not yet included in the published data.

Existing PE data will be complemented by a new derived dataset, which estimates actual evapotranspiration (ET) as the residual term from measurements of net radiation, soil heat flux and the sensible heat flux derived from sonic anemometer measurements. Modelled energy fluxes, such as latent and sensible heat, have been calculated by utilising the 20 Hz wind
measurements recorded at the majority of COSMOS-UK sites (Crowhurst et al., 2019). This provides a network-wide modelled actual ET dataset for the UK.

In addition to the measurements mentioned previously, COSMOS-UK sites also capture photographs. Sites include a camera for monitoring phenology, a 'PhenoCam', with two hemispheric lenses facing north and south (Fig. 9). Each COSMOS-UK site sends five photographs per day, which capture the full extent of the COSMOS-UK site and surrounding area, thereby
providing additional information on local phenology and cloud cover. These PhenoCam images can be used to confirm when site conditions have changed, for example when the land cover has been modified (e.g. ploughing, mowing, grazing, harvesting) or there has been heavy snowfall. PhenoCam photos from COSMOS-UK sites are also currently being analysed to produce a greenness dataset. Using site-specific image masks, RGB (red, green, blue) data can be extracted from each image to determine the greenness of the land cover at each site (Wingate et al., 2015). In 2020 the network's first gauge board was
installed at the Cwm Garw site in Wales. Gauge boards indicate height above ground level (cm) against which vegetation height and snow depth can be estimated via PhenoCam images. Further gauge boards are planned at sites across the network.

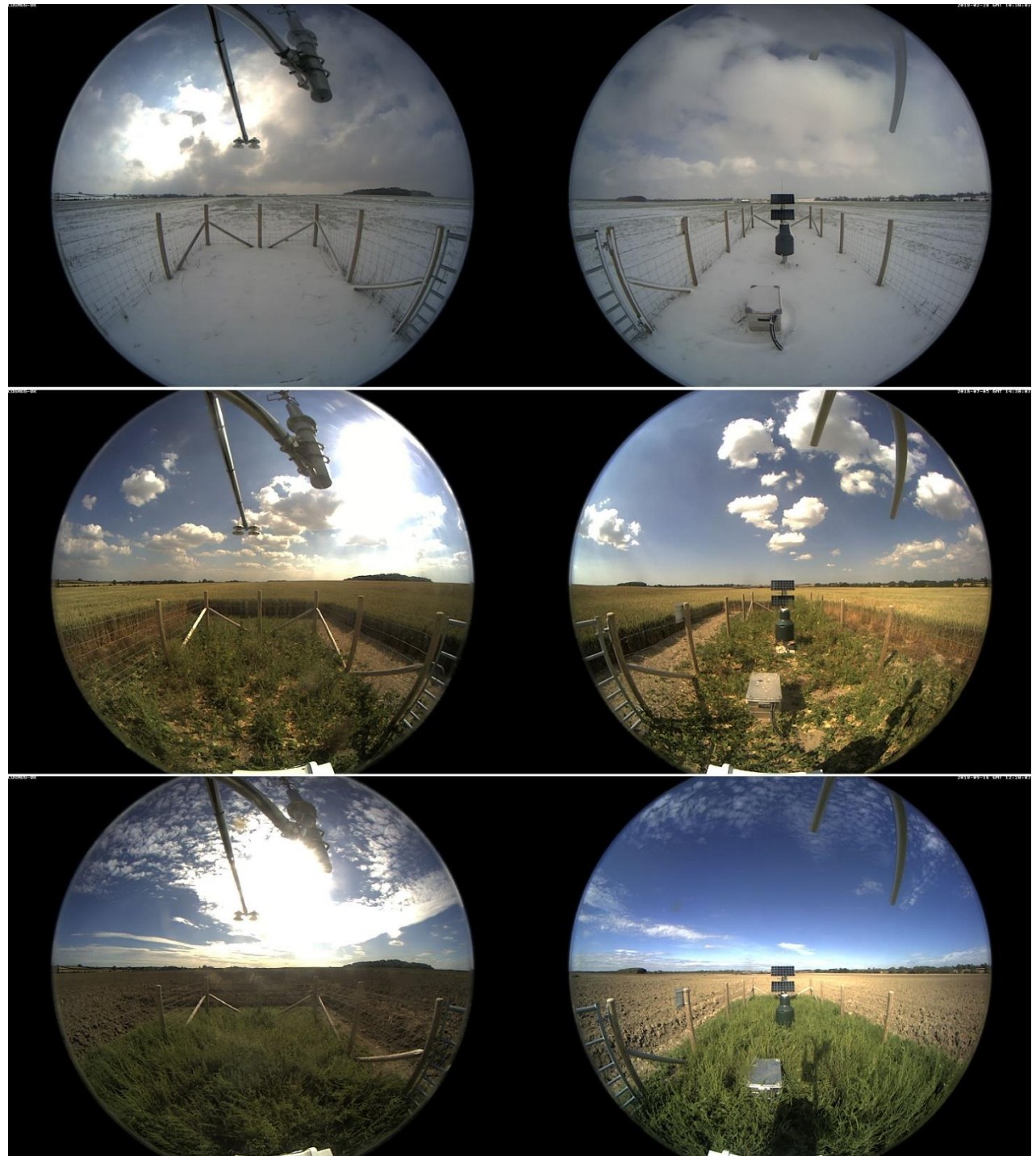

**Figure 9: PhenoCam photographs from the Fincham COSMOS-UK site in East Anglia. From top: a snow event at the end of February 2018; oil seed rape crop growing in surrounding field in July 2018; and the bare field in September 2018.**

## 4 Data availability

The "Daily and sub-daily hydrometeorological and soil data (2013-2019) [COSMOS-UK]" time series dataset is the most recent COSMOS-UK dataset at the date of publication. The dataset is published by, and available for download from, the

EIDC at https://doi.org/10.5285/b5c190e4-e35d-40ea-8fbe-598da03a1185 (Stanley et al., 2021).

This dataset comprises daily and sub-daily observations and derived data between 02 October 2013 and 31 December 2019 inclusively for 51 sites across the UK. The files included for each site are:

1. COSMOS-UK_[SITE_ID]_HydroSoil_SH_2013-2019.csv
2. COSMOS-UK_[SITE_ID]_HydroSoil_SH _2013-2019_QC_Flags.csv

3. COSMOS-UK_[SITE_ID]_HydroSoil_Hourly_2013-2019.csv
4. COSMOS-UK_[SITE_ID]_HydroSoil_Daily_2013-2019.csv

Table 7 comprises the measured and derived variables, units and temporal resolution of data available in these files. File 1 contains measured and derived variables at 30 minute resolution and file 2 comprises the QC flags for the data in file 1. File 3 comprises the derived variables available at hourly resolution and file 4 contains derived data at daily resolution.


Table 7: Measured and derived variables available in the 4 data files provided in the COSMOS-UK dataset (Stanley et al., 2021).

| Variable | Unit | Data type | Data resolution | File |
|---|---|---|---|---|
| Precipitation | mm | Measured | 30 min | 1 |
| Relative humidity | % | Measured | 30 min | 1 |
| Absolute humidity | $gm^{-3}$ | Derived | 30 min | 1 |
| Air temperature | °C | Measured | 30 min | 1 |
| Atmospheric pressure | hPa | Measured | 30 min | 1 |
| Incoming longwave radiation | $Wm^{-2}$ | Measured | 30 min | 1 |
| Incoming shortwave radiation | $Wm^{-2}$ | Measured | 30 min | 1 |
| Outgoing longwave radiation | $Wm^{-2}$ | Measured | 30 min | 1 |
| Outgoing shortwave radiation | $Wm^{-2}$ | Measured | 30 min | 1 |
| Net radiation | $Wm^{-2}$ | Derived | 30 min | 1 |
| Wind direction | degrees | Measured | 30 min | 1 |
| Wind speed | $ms^{-1}$ | Measured | 30 min | 1 |
| 3D wind speed data (x3) | $ms^{-1}$ | Measured | 30 min | 1 |
| Snow depth | mm | Measured | 30 min | 1 |
| Soil heat flux (x2) | $Wm^{-2}$ | Measured | 30 min | 1 |
| Soil temperature (x5) | °C | Measured | 30 min | 1 |
| Soil temperature (TDT) (x2 or x10) | °C | Measured | 30 min | 1 |

| | | | | |
|---|---|---|---|---|
| Soil moisture (TDT VWC) (x2 or x10) | % | Measured | 30 min | 1 |
| Soil moisture (CRNS VWC) | % | Derived | Hourly & Daily | 3 & 4 |
| Effective depth of CRNS (D86 at 75 m) | cm | Derived | Hourly & Daily | 3 & 4 |
| Neutron counts from CRNS (corrected) | counts | Derived | Hourly | 3 |
| Potential evaporation | mm | Derived | Daily | 4 |
| Albedo | Dimensionless | Derived | Daily | 4 |
| Snow days | Yes/No | Derived | Daily | 4 |
| Snow Water Equivalent (from CRNS) | mm | Derived | Daily | 4 |

Site metadata are available in four additional files:

    5.   COSMOS-UK_SiteMetadata_2013-2019.csv

6.   COSMOS-UK_HydroSoil_SH_2013-2019_Metadata.csv

    7.   COSMOS-UK_HydroSoil_Hourly_2013-2019_Metadata.csv

    8.   COSMOS-UK_HydroSoil_Daily_2013-2019_Metadata.csv

Data availability for individual variables and sites varies throughout the dataset due to sensor faults, planned preventative maintenance, and disruptions to data collection. Overall data completeness for this period for available variables is 95.5% (see

a summary in Fig. 10) (Stanley et al., 2021). Missing values due to technical faults and failed QC calculations are recorded as -9999.

COSMOS-UK has been designed as a long-term monitoring network and further data will be made available via the EIDC. The dataset is superseded annually, with the inclusion of one additional year of COSMOS-UK data for all available sites. Data are provisional and subject to change with the release of each new version in line with developments to the science,

instrumentation, data processing, quality control, and data gap-filling protocols. Data are supplied with supporting information and a data licence that outlines the terms of use to data users.

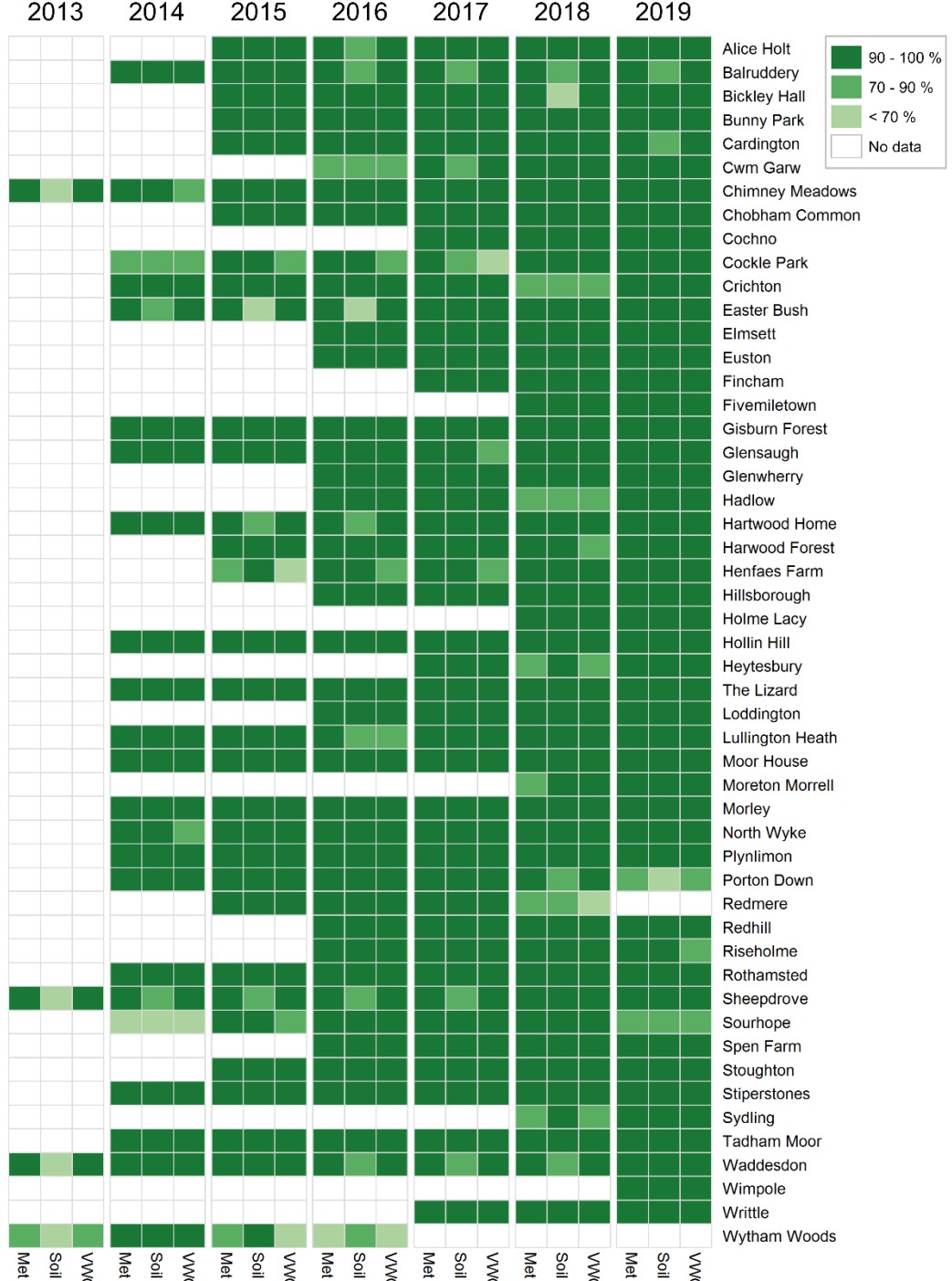

**Figure 10: Data completeness for the Stanley et al. (2021) COSMOS-UK dataset. 'VWC' is the CRNS VWC data; 'Soil' consists of data from buried point and profile soil moisture sensors; and 'Met' comprises meteorological variables.**

# 5 Data applications

Observational data from the COSMOS-UK network have been used for a variety of purposes. They have significant potential to empower a range of existing and novel scientific applications. Descriptions of some uses are included in this section. The main and immediate applications for COSMOS-UK observational data are for use in hydrological and land-surface models
and for validating remote sensing data.

COSMOS-UK measurements cover a range of environmental characteristics and this can be exploited for development of models, which are used for scaling up and forecasting soil moisture at the national scale. Field scale soil moisture measurements from a variety of land covers have been used to investigate the accuracy and reliability of LSMs. Comparison of COSMOS-UK soil moisture measurements with outputs from LSMs allows for investigation into those models' ability to represent soil
moisture dynamics and underlying physical processes (Cooper et al., 2020a). For example, data assimilation techniques have been used to adjust soil physics parameters (via pedo-transfer functions), thereby allowing the JULES model to more closely produce the observed range of soil moisture values (Cooper et al., 2020b). This demonstrates the value in using in situ COSMOS-UK data to drive models for increased performance. Additional potential exists in using these larger area data across a variety of land covers to explore interactions and dynamics in infiltration, run-off (Dimitrova-Petrova et al., 2020) and
interception (Zreda et al., 2012). Improved understanding of these processes could lead to more accurate and reliable modelling of, and thus improved forecasting for, a range of hydrological phenomena. For instance the JULES model, used as the land-surface scheme in UK Met Office forecasts (Best et al., 2011), is run at a minimum scale of 1 km. The parameterisation of this model can be improved in response to these soil moisture data (Cooper et al., 2020b), which can then be used with UK scale meteorological data (Robinson et al., 2020) to deliver a national scale soil moisture product.

Using land-atmosphere modelling together with COSMOS-UK soil moisture and modelled ET data, along with measured ET where available, can empower further investigation into soil moisture dynamics and biosphere-atmosphere fluxes. These combined data can provide greater understanding of land-atmosphere processes, for example of feedback events during periods of drying soils and extreme air temperatures (Dirmeyer et al., 2021) and storm initiation (Taylor et al., 2012). Use of these data can also help estimate landscape average precipitation, as described in Franz et al. (2020).

COSMOS-UK field scale soil moisture is also proving particularly useful for ground-truthing remote sensing soil moisture data. For this application, the value of COSMOS-UK data largely resides in the footprint of the CRNS. The field scale soil moisture data prove to be a radical improvement on point soil measurements alone, as the larger footprint more closely represents the resolution of satellite products, whilst averaging across smaller-scale soil heterogeneity. COSMOS-UK data can therefore help validate and improve existing products (Beale et al., 2020; Pinnington et al., 2020; Quinn et al., 2020) for
obtaining better estimates of UK soil moisture data at higher spatial resolution (Peng et al., 2020). Similar networks across the globe, for example in the US, India and China, have also been exploited for such research (Montzka et al., 2017; Upadhyaya et al., 2021; Zhu et al., 2019). COSMOS-UK soil moisture can be used together with PhenoCam data to further investigate remote sensing analysis in vegetation growth, crop senescence, snow events, surface ponding, and land cover change.

With a vision to develop a dynamic near-real time UK soil moisture map, there is potential for COSMOS-UK data to influence wider fields. Scaled up near-real time COSMOS-UK data either through using models, remote sensing, or both could inform water-regulators such as the Environment Agency on the state of UK soil moisture. Direct evidence of drought and flooding events induced, or impacted, by soil moisture is increasingly needed to inform decisions at national scale. Similarly, these data could help inform UK wildfire prediction and ecological applications via simulations of soil moisture, air temperature, precipitation, and vegetation information (Albertson et al., 2009). Additionally, with an understanding of the links between soil moisture and plant productivity, COSMOS-UK data can be used to monitor the need for irrigation (Ragab et al., 2017), thereby improving our predictions of crop yield for the UK. Furthermore, understanding soil moisture at identified landslip sites could help in the development of Landslide Early Warning systems, for example using the Hollin Hill COSMOS-UK site in North Yorkshire (Bliss et al., 2020). At site scale, soil moisture data from individual COSMOS-UK sites have proven valuable when paired with gas flux data provided by field scale methodologies such as eddy covariance (Cowan et al., 2018, 2020). Here the high temporal, spatially-integrated soil moisture data can be used to better refine gap-filling methods, particularly for emissions of the powerful GHG nitrous oxide, which responds strongly to changes in soil aerobicity. As all of the major GHGs ($CO_2$, $CH_4$, $N_2O$), and many secondary GHGs and other sources of air pollution (CO, NO, $NO_2$) generated by soil microbial activity, are heavily influenced by soil moisture (Cowan et al., 2018; Davidson et al., 2000; Oertel et al., 2016; Van Den Pol-van Dasselaar et al., 1998), the COSMOS-UK network will provide the ability to better refine UK scale emission inventories in the future as UK scale soil moisture models are improved.

COSMOS-UK data could also provide insight into alternative scientific research, such as the relationship between soil moisture and pest behaviour (Hertl et al., 2001); the impact of soil moisture on local infrastructure (Pritchard et al., 2013); investigation of ground level cosmic ray events (Flückiger et al., 2005); and meteorological data with respect to bacterial infection seasonality (Djennad et al., 2019).

## 6 Conclusions

The COSMOS-UK network is the world's most spatially dense national network of cosmic-ray neutron sensors for observing near-surface field scale soil water dynamics. Field scale soil moisture and hydrometeorological data are available from a diverse range of sites located across the UK, with the earliest sites providing data since 2013. The COSMOS-UK dataset is a unique and growing resource that has already captured soil water dynamics across a wide range of climatic conditions, including extreme events such as the extended cold wave, heatwave and agricultural drought the UK experienced during 2018. As the length of the data record continues to grow, COSMOS-UK will provide an unprecedented resource for national scale environmental monitoring. Data from the COSMOS-UK network are of significant national and international relevance empowering applications including the validation of remotely sensed data products, the interpretation of biogeochemical flux observations, and the calibration and testing of LSMs. Significant opportunity exists for new applications in support of water resources, weather prediction and space sciences, and biodiversity and environmental change.

At the time of publication, the most recent COSMOS-UK dataset available comprises daily and sub-daily hydrometeorological and soil physics data between 02 October 2013 and 31 December 2019 for 51 sites. The COSMOS-UK dataset will be updated on an annual basis.


## Author contributions

The initial draft manuscript was prepared and written by HC with significant contributions from JB, DB and SS. Additional manuscript contributions were made by E Bennett, E Blyth, EC, NC, JE, MF, AJ, RM, DR, MS and ET. AA, JB, MB, MC, HC, NC, AC, JE, PF, WL, RM, DR, PS, JT, ET, AW and BW conducted fieldwork and maintained network operation. JB, E
Blyth, DB, EC, HC, JE, MF, OH, RM, SS, MS and ET contributed to data processing, quality control and applications. VA, E Bennett, DB, OH, SS and OS contributed to data accessibility. LB, E Bennett, DB, JE and AJ contributed to the planning and management of the network. All authors contributed to the success of the dataset.

## Competing interests

The authors declare that they have no conflict of interest.

## Acknowledgements

The authors would like to thank past and present COSMOS-UK team members for their support of the monitoring network: Joshua Alton, Sarah Bagnoli, Sandie Clemas, Louisa Doughty, Richard Ellis, Charles George, Duncan Harvey, Ned Hewitt,
Filip Kral, Sarah Leeson, Jeremy Libre, Gemma Nash, Joanne Newcomb, Matthew Parkes, Ragab Ragab, Warren Read, Colin Roberts, Ondrej Sebek, Andrew Singer, Charlie Stratford, Simon Teagle, Helen Vincent, John Wallbank, Helen Ward and George Wright.

This work was supported by the Natural Environment Research Council award number NE/R016429/1 as part of the UK-SCAPE programme delivering National Capability.
The authors would like to thank the Physikalisches Institut, University of Bern, Switzerland for kindly and reliably providing Jungfraujoch neutron monitor data.

The authors also thank the University of Delaware Department of Physics and Astronomy and the Bartol Research Institute, USA for the reliable provision of Newark/Swarthmore neutron monitor data.

The authors would also like to thank Polar Geophysical Institute Russian Academy of Sciences, Russia for reliably providing
data from the Apatity neutron monitor.

The authors thank the organisations and individual site owners who have provided the land, assistance and access for each of our monitoring sites.

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
