# Peer review of "COSMOS-UK: National soil moisture and hydrometeorology data for environmental science research"

_Earth System Science Data, 2020_

## Referee Comment (RC1) · Anonymous Referee #1 · 1 Dec 2020

The work describes the cosmic-ray soil moisture observation network of the United Kingdom, which has been built up over the past 7 years. As stated by the authors, the network is indeed the densest national network of such kind and offers in combination with its complementing climate observations a valuable data set for environmental research. In the paper the focus is mainly on the description of the network, its equipment and the local calibration to relate the neutron counts to volumetric soil moisture. I think, both, the data set and the data description paper are relevant and important to environmental modeling and help to improve our understanding of land-surface – atmosphere interactions. Nevertheless, the documentation needs some further clarifications and also the structure and content of the data set should be partially reworked. Therefore,

[Figure]

I recommend the publication, after the consideration of the following comments:

In general, the manuscript is well structured but I see shortcomings in two respects:

1) I think the data description is missing information about and a discussion of possible errors within the CRNS method / data 2) some of the essential data processing steps are not explained with sufficient detail

More specifically my comments are:

1: (Title) "empowering UK environmental science" To me this formulation reads a bit "selfish". It may be UK-national soil moisture data, but it should be (and I belief it is) open to anyone who works in the field of environmental science, British or not. I think there are many global or continental models that include the UK and would likewise benefit from your data. So I suggest to change the title into something less "nationalistic".

35: Could you find another citation Moene and van Dam? I think there are plenty of papers out there that give a more specific introduction to the topic of soil moisture in the Earth System, e.g., by Seneviratne et al.

40: "measurement footprint" You may already specify it here to let the reader know what to expect. E.g. "the field scale measurement footprint"

57: 80cm -> blank missing

80-85: What's the reason for not having more stations in Scotland (only 2 stations in the east)? Also for Yorkshire and North-West England the network is more sparse.

Table 1: I think the table needs to be condensed. Since available in the metadata, you can skip the 2 columns for Easting and Northing and also End date can be moved to Start date like this: "Start(end) date" 26/11/2013(-01/10/2016)

The numbers for SAAR and Altitude should be right-adjusted. For the Soil type and land cover you should define abbreviations in the table header (e.g. MS for Mineral

soil, IG for Improved grassland). and Soil type should go right of Altitude and SAAR. So finally one line could be a s short as: Cochno 23/08/2017 168 662 MS IG

and all stations could be overviewed at a glimpse one a single page.

120-122: not clear if all CRS1000B sensors have been removed from the network and CRS2000Bs are used now everywhere. If all had been changed, since when is the network pure CRS 2000B? Pls. be more specific.

125: Shortly explain what TDT sensors are.

127: What was the reason to remove the PICO profiles?

139: Why don't you use wind shields for the Pluvios? It's so often windy in Britain.

157: Pls. list the 8 sites with extra snow measurements

169: Is there any publication that shows the superiority of the SBS500, so that you want to use it as reference to check the quality of others (like the Pluvio)?

183-184: Why do you only consider 0-25 cm depth for calibration? Later in figure 7 you suggest much larger penetration depths.

183: Pls. provide model and manufacturer for the soil augers

189: Fig.3: You should also provide annuluses (25,75, 200m) for the older calibrations (e.g. by having 2 separate graphs side by side with one legend, or by additional annotation in brackets '5 (25) m')

205: Were the bulk densities also obtained with the weighting function?

206: Did you also consider the findings of Schroen et al. 2017 https://hess.copernicus.org/articles/21/5009/2017/ for your weighting and calibration approach? The Sheepdrove data of COSMOS-UK was used in this work.

208-210: There should be a discussion about the magnitude of possible errors in the determination of the reference soil moisture. Is the 0.03 vol.% difference significant or

may the error in sampling and thermo-gravimetry be even higher?

211: Tab. 5: Please give standard deviations for bulk density, lattice water, and soil organic carbon (add with $\pm$ sign)

217-230: Please be more specific in the description of the derivation of the corrected neutron count signal. This important step should be reproducible by the data users. So you should provide the exact formulas and specify the constants used (e.g. for the reference pressure). Did you use the same reference pressure for all stations? Did you do or do you plan to do a cross-calibration for the different sites, to get an idea of the variations in sensitivity of the sensors?

224: I think it should be "Physikalisches Institut's, University of Bern, Jungfraujoch"

227: What are the implications of using a neutron monitor with such a large displacement and elevation difference. I assume a big difference in the cutoff rigidity between Jungfraujoch and your sites. Wouldn't the Kiel monitor be better suited? I think that this should also be discussed in the paper.

241: Are the fits for all the sites performing equally well? You may discuss and add some other cases too.

243: Fig. 4: Please add the formula for the synthetic curve including constants to the legend of the plot

264: Fig. 5: Please add panel IDs (a, b, c). What is the reason for the counts in a) being half of those in b)? Is it a different aggregation interval? X-axes text (date) should be removed for all but the lowermost panel. Then the margins between panels can be removed in favor of larger legend font sizes. Y-axis text for e) Precipitation is too small (smaller than for other panels). Legends (font sizes) for d) VWC are too small.

271: Fig. 6: Please add panel IDs. Resolution of figure needs to be improved. Since colors are rather hard to distinguish (especially with red-green blindness), I would recommend to use gray-scale (maybe with transparency) for the lines and symbols (point,

cross, dot, ...) for the regions at the highlighted date.

289: "measurements" I wouldn't see this as a measurement but rather as a derived quantity. Something of "shows the assumed/computed/estimated D86..."

293: Fig. 7: Add panel IDs. All texts are by far too small. As for Fig. 5 I suggest to remove the x-axes annotation and to cut the margin between the sub-panels. How can the Euston VWC time-series reach 0? Was that reflected by the TDT sensors?

328: As you provided SnowFox derived SWE in the data set, its derivation should be documented more precisely (formula, calibration). At least provide it as supplementary material.

334: Fig. 8: Same as before (panel IDs and description in the caption), remove x-axes between sub-panels, increase font sizes and avoid red-green coloring in one graph.

353: Please explain what "gauge boards" are

361-386: The different data sub-products are clearly distinguished and organized in a logical structure. However, on top-level, the user doesn't want to find a 200 files long list to be downloadable click by click. So all the data should be organized as a single file archive (zip) download.

The user should be able to use the data set without considering the data-description paper just by making use of the metadata.

Thus, the information in table 7 should be provided within 4 additional metadata files (SH, SH...QC, Hourly, Daily). And the JSON format might be a better way to specify things. What's lacking in the metadata is the information how a certain measurement is derived does the timestamp 00 refer to the period 00-30 minutes or 30-00 minutes and has the value been obtained by averaging or as instantaneous value?

The timestamp in the data files need to be ISO 8601 compliant (e.g. 2018-01-01T12:00:00Z) so that also the time-zone information is contained.

The site metadata should also contain the standard deviations for BD, SOC, and lattice water (which is btw. completely missing). For the easting and northing the projection needs to be specified.

Please provide also the hourly raw neutron counts (uncorrected) as well as the Snow-Fox raw and corrected counts as some people may be interested in using their own corrections.

387: Fig. 10: Change to color-blind safe palette (https://knightlab.northwestern.edu/2016/07/18/three-tools-to-help-you-make-colorblind-friendly-graphics/)

435-447: It would be nice to have also a short outlook. What are the perspectives for COSMOS-UK? Are there plans to extend the network (e.g., for Scotland)? How long is the projected lifetime of the project. How will the CRNS sensors degrade over time? Do you plan to upgrade the network with more sensitive detectors as they become available on the market?

---

## Referee Comment (RC2) · Anonymous Referee #2 · 1 Dec 2020

The manuscript presents the data details of the COSMOS-UK network of soil moisture sensors. This is most certainly a unique network of soil moisture (and related variables) sensors; the research community will greatly benefit from the details provided in the manuscript. The manuscript is well written and makes an interesting reading. I recommend acceptance of the manuscript for publication subject to minor revisions arising from the following comments:

1. The main issue is with an absence of calibration details : Table 5 and the discussion preceding it provide elaborate details of the sampling procedure and determination of soil moisture in the lab. The effectiveness of calibration is however mentioned in just

one sentence, "There was < 0.03 cm3 cm-3 difference in VWC between the soil moisture determined from these samples and the corresponding daily VWC value derived using the site's initial calibration data". Table 5 only shows the reference values and not the difference between the point scale measurements and the CRNS data. In the absence of such results, it is difficult to judge how good has the calibration been. Perhaps in the supplementary material, more detailed results of calibration could be given.

2. The claim that the spatial data on soil moisture leads to improved hydrometeorological forecasts needs to be substantiated, either by citing appropriate references or through a convincing argument.

3. Table 1 gives details of the COSMOS-UK stations. Please consider highlighting the mountainous sites, if any among these. The altitude values shown in the table do indicate the heights at which the stations are located, but a higher altitude does not necessarily indicate a station on a mountain. It would also be interesting to see the soil moisture signatures of the mountainous regions as compared to those in the plains.

4. Is there any irrigation in the area around the CRNS stations? If yes, how has it affected the soil moisture data? Is it possible to filter out the effect of irrigation in the data?

5. Table 4 shows that at the same depth two TDT point source sensors are located. How are they spaced? How is the data from the two sensors merged?

6. Lines 183-184 : Volumetric samples are taken at five depths, upto 0.25 m bgl. However, the CRNS data is between 0.1 m to 0.8 m. Is there any calibration for depths below 0.25 m?

7. Fig 2 (appearing around line 272) should be Fig. 6.

8. It is difficult to understand the "automatic processing" for quality checks. Table 6 provides the flags raised for various events, but how these events are identified in the data is not clear (for example, how does the automatic processing detect simultaneously,

missing data and small sample of data?)

9. Fig. 8 caption may be made more descriptive. Also, the caption states that these are the observations required for PE calculation, but the last panel in fact shows the PE calculated! It would also be interesting to see the soil temperature plot along with the other variables shown here (although soil temperature is not used in PE calculation).

10. Please also discuss how the soil moisture measured at these 51 stations may be smoothened to upscale it to the national scale (see line no. 396).

---

## Author Comment (AC1) · 21 Jan 2021

The authors thank both referees for providing detailed and helpful comments, suggestions and questions. We will add detail and clarity to the manuscript where suggested.

With regard to comments concerning the inclusion of individual calculations and more detailed explanations of variable derivation and methodology, we believe the manuscript describes the data set at an appropriate level of detail for the intended end users who are interested in soil moisture and hydrometeorology measurements. This description paper is not aimed at researchers looking to further develop the cosmic-ray method of soil moisture measurement.

[Figure]

The COSMOS-UK team is in active discussion with researchers across the European COSMOS community, with the intention of producing a more detailed output to describe specific methodologies for deriving soil moisture from the CRNS.

We address each of your individual comments (RC) with our response (AR) below.

Response to Referee 1

RC1 1: (Title) "empowering UK environmental science" To me this formulation reads a bit "selfish". It may be UK-national soil moisture data, but it should be (and I belief it is) open to anyone who works in the field of environmental science, British or not. I think there are many global or continental models that include the UK and would likewise benefit from your data. So I suggest to change the title into something less "nationalistic".

AR1 We had no intention for the title to imply any such meaning and we will therefore carefully amend it. The revised title will read, "COSMOS-UK: National soil moisture and hydrometeorology data for environmental science research".

RC2 35: Could you find another citation Moene and van Dam? I think there are plenty of papers out there that give a more specific introduction to the topic of soil moisture in the Earth System, e.g., by Seneviratne et al.

AR2 Thank you for your suggestion, we will reference the work by Seneviratne et al. (2010).

RC3 40: "measurement footprint" You may already specify it here to let the reader know what to expect. E.g. "the field scale measurement footprint"

AR3 We will amend the text to "field scale measurement footprint".

RC4 57: 80cm -> blank missing

AR4 This will be corrected.

[Figure]

RC5 80-85: What's the reason for not having more stations in Scotland (only 2 stations in the east)? Also for Yorkshire and North-West England the network is more sparse.

AR5 Installation of COSMOS-UK sites was initially focussed in areas where soil moisture was expected to have greater variability and where practical and logistical challenges related to access and communication were minimised. Installation of sites in less-represented regions is in consideration but is dependent on the availability of resources. This will be made clearer in the manuscript.

RC6 Table 1: I think the table needs to be condensed. Since available in the metadata, you can skip the 2 columns for Easting and Northing and also End date can be moved to Start date like this: "Start(end) date" 26/11/2013(-01/10/2016) The numbers for SAAR and Altitude should be right-adjusted. For the Soil type and land cover you should define abbreviations in the table header (e.g. MS for Mineral soil, IG for Improved grassland). and Soil type should go right of Altitude and SAAR. So finally one line could be a s short as: Cochno 23/08/2017 168 662 MS IG and all stations could be overviewed at a glimpse one a single page.

AR6 Thank you for this suggestion for Table 1. We agree that this will greatly improve the table and we will amend it as suggested.

RC7 120-122: not clear if all CRS1000B sensors have been removed from the network and CRS2000Bs are used now everywhere. If all had been changed, since when is the network pure CRS 2000B? Pls. be more specific.

AR7 The text will be revised for clarity. Only the first four sites were installed with a CRS1000B (Chimney Meadows, Sheepdrove, Waddesdon and Wytham Woods), whilst all other sites were installed with a CRS2000B sensor. Wytham Woods was decommissioned in October 2016. In February 2020, CRS2000B sensors were installed at Chimney Meadows, Sheepdrove and Waddesdon alongside the existing CRS1000B sensors which remain connected.

RC8 125: Shortly explain what TDT sensors are.

AR8 The following detail will be added, "Each site includes point scale soil moisture sensors, which estimate VWC via Time Domain Transmissometry (TDT). These TDT sensors estimate point scale soil moisture by measuring the time taken for an electromagnetic wave to travel along the sensor's closed circuit; this transmission decreases in speed with soil permittivity (Blonquist et al., 2005)."

RC9 127: What was the reason to remove the PICO profiles?

AR9 We experienced issues with this sensor as configured at our sites. Specifically the configuration resulted in a high failure rate for other sensors and resulted in loss of data. Because of this, these instruments were removed during 2019 - 2020 in order to maximise the data capture across the array of instruments. This will be made clearer in the manuscript.

RC10 139: Why don't you use wind shields for the Pluvios? It's so often windy in Britain.

AR10 Pluvio wind protection shields were omitted during COSMOS-UK site installations as site locations were identified as being not particularly exposed. It is acknowledged that this decision could impact the extent of wind-induced under-catch, and shields will be considered during future network improvements. This will be made clear in the manuscript.

RC11 157: Pls. list the 8 sites with extra snow measurements

AR11 We will amend the text to read, "These sites (Glensaugh, Easter Bush, Gisburn Forest, Plynlimon, Sourhope, Moor House, Cwm Garw and Cochno) were installed with...".

RC12 169: Is there any publication that shows the superiority of the SBS500, so that you want to use it as reference to check the quality of others (like the Pluvio)?

AR12 We will amend the text to include relevant citations. The design of the EML SBS500 has improved aerodynamic characteristics (Strangeways, 2004), and Colli et al. (2018) report that when compared with other assessed rain gauges, the SBS500 exhibits superior reduction in turbulence and under catch.

RC13 183-184: Why do you only consider 0-25 cm depth for calibration? Later in figure 7 you suggest much larger penetration depths.

AR13 Soil sampling depths were selected to match typical (moist) UK conditions. Ideally, subject to cost, calibrations should be repeated at different VWCs, preferably capturing the full range of VWC measured at a particular site. We will add clarity to the text and highlight that the shallow soil layers are given higher weighting than deeper layers to represent the decreasing contribution of deeper soil water (Köhli et al., 2015; Schrön et al., 2017). Figure 7 will be amended; this currently shows incorrect (higher) values for D86. We apologise for this error and we will correct the data set accordingly.

RC14 183: Pls. provide model and manufacturer for the soil augers

AR14 We will include these details.

RC15 189: Fig.3: You should also provide annuluses (25,75, 200m) for the older calibrations (e.g. by having 2 separate graphs side by side with one legend, or by additional annotation in brackets '5 (25) m')

AR15 We will amend Figure 3 to include past calibration distances.

RC16 205: Were the bulk densities also obtained with the weighting function?

AR16 We do not currently obtain the bulk densities with the weighting function, however we will consider this in future work. This will be made clear in the manuscript.

RC17 206: Did you also consider the findings of Schroen et al. 2017 https://hess.copernicus.org/articles/21/5009/2017/ for your weighting and calibration approach? The Sheepdrove data of COSMOS-UK was used in this work.

AR17 Yes these findings have been considered in our approach; when applying the soil sampling calibration, the shallow 5 cm layers are given a higher weighting than the deeper ones. Please note that the Sheepdrove COSMOS-UK site is unrelated and in a different location to the Sheepdrove monitoring station referred to by Schrön et al. (2017). We will make this clearer in the manuscript.

RC18 208-210: There should be a discussion about the magnitude of possible errors in the determination of the reference soil moisture. Is the 0.03 vol.% difference significant or may the error in sampling and thermo-gravimetry be even higher?

AR18 This is an interesting question which we are unable to answer fully at this time, however we will add detail and address this as part of future recalibrations.

RC19 211: Tab. 5: Please give standard deviations for bulk density, lattice water, and soil organic carbon (add with $\pm$ sign)

AR19 We agree these are needed and we will add these to Table 5.

RC20 217-230: Please be more specific in the description of the derivation of the corrected neutron count signal. This important step should be reproducible by the data users. So you should provide the exact formulas and specify the constants used (e.g. for the reference pressure). Did you use the same reference pressure for all stations? Did you do or do you plan to do a cross-calibration for the different sites, to get an idea of the variations in sensitivity of the sensors?

AR20 After consideration we believe that the current description with relevant citations provides an acceptable level of detail for the data set's intended users, who are interested in soil moisture and other environmental data. We will add clarity regarding reference pressure; a value of 1000 hPa is used for all sites, whilst the instantaneous barometric attenuation length is calculated for each site. A selection of CRS2000/B sensors deployed at COSMOS-UK sites were cross-compared for sensitivity under lab conditions prior to installation. Where necessary, in situ sensitivity comparisons have

been completed by collecting several months of data from adjacent sensors.

RC21 224: I think it should be "Physikalisches Institut's, University of Bern, Jungfraujoch"

AR21 Thank you, we will correct this text.

RC22 227: What are the implications of using a neutron monitor with such a large displacement and elevation difference. I assume a big difference in the cutoff rigidity between Jungfraujoch and your sites. Wouldn't the Kiel monitor be better suited? I think that this should also be discussed in the paper.

AR22 We will add detail and discussion to the manuscript. We use normalised count rates in the intensity correction, which are not greatly affected by their cut-off rigidity in the absence of significant space weather events. During such events, the cut-off rigidity of a specific location may change due to magnetic field disturbances, so only matching cut-off rigidity might not compensate for such events. When comparing Jungfraujoch neutron monitor with the available monitors of similar cut-off rigidities to the COSMOS-UK sites, the normalised counts and the associated trends were in good agreement. When choosing the most suitable neutron monitor for this work, Jungfraujoch was identified as a well-maintained monitor with a high level of data completeness, however we will continue to monitor research in this area to improve our methods where possible. The Kiel monitor is currently described as not having efficiency-corrected values available, so we will reconsider this monitor when these data are available.

RC23 241: Are the fits for all the sites performing equally well? You may discuss and add some other cases too.

AR23 There may be some misunderstanding with this question. This figure does not illustrate goodness of fit, it only illustrates the portion of the calibration curve corresponding to the observed count rates. We will make this clearer in the manuscript and add some other cases.
RC24 243: Fig. 4: Please add the formula for the synthetic curve including constants to the legend of the plot

AR24 The formula will be included.

RC25 264: Fig. 5: Please add panel IDs (a, b, c). What is the reason for the counts in a) being half of those in b)? Is it a different aggregation interval? X-axes text (date) should be removed for all but the lowermost panel. Then the margins between panels can be removed in favor of larger legend font sizes. Y-axis text for e) Precipitation is too small (smaller than for other panels). Legends (font sizes) for d) VWC are too small.

AR25 Thank you for your suggestions and spotting an error in the plot. Panel IDs a, b, c will be added to Figure 5. The data in panel a will be replaced with the intended aggregation interval. Your other suggestions for improvement will also be included.

RC26 271: Fig. 6: Please add panel IDs. Resolution of figure needs to be improved. Since colors are rather hard to distinguish (especially with red-green blindness), I would recommend to use gray-scale (maybe with transparency) for the lines and symbols (point, cross, dot, ...) for the regions at the highlighted date.

AR26 We will add panel IDs to Figure 6, improve the resolution and make the different groups more distinguishable for all users.

RC27 289: "measurements" I wouldn't see this as a measurement but rather as a derived quantity. Something of "shows the assumed/computed/estimated D86..."

AR27 We will change the text to read, "...shows the estimated D86 values for a...".

RC28 293: Fig. 7: Add panel IDs. All texts are by far too small. As for Fig. 5 I suggest to remove the x-axes annotation and to cut the margin between the sub-panels. How can the Euston VWC time-series reach 0? Was that reflected by the TDT sensors?

AR28 Figure 7 will be amended as suggested. The Euston site is located on very well-draining sandy soil and during the extreme dry event in 2018 the derived soil moisture

data were consistently low. The minimum daily average TDT VWC readings between 2018-05-01 and 2018-08-31 were: 1.3, 0.7, 2.3, 0.5, 2.7, 1.6, 2.2, 0.2, 0.3 and 0.4 at the respective depths of 5, 5, 10, 10, 15, 15, 25, 25, 50 and 50 cm.

RC29 328: As you provided SnowFox derived SWE in the data set, its derivation should be documented more precisely (formula, calibration). At least provide it as supplementary material.

AR29 The data set only provides SWE derived from the CRNS (not the SnowFox). We will add clarity and additional detail alongside the citation.

RC30 334: Fig. 8: Same as before (panel IDs and description in the caption), remove x-axes between sub-panels, increase font sizes and avoid red-green coloring in one graph.

AR30 Figure 8 will be amended with your suggestions.

RC31 353: Please explain what "gauge boards" are

AR31 We will amend the manuscript to include additional detail, "In 2020 the network's first gauge board was installed at the Cwm Garw site in Wales. Gauge boards indicate height above ground level (in cm) against which vegetation height and snow depth can be estimated via PhenoCam images. Further gauge boards are planned at sites across the network."

RC32 361-386: The different data sub-products are clearly distinguished and organized in a logical structure. However, on top-level, the user doesn't want to find a 200 files long list to be downloadable click by click. So all the data should be organized as a single file archive (zip) download. The user should be able to use the data set without considering the data-description paper just by making use of the metadata. Thus, the information in table 7 should be provided within 4 additional metadata files (SH, SH...QC, Hourly, Daily). And the JSON format might be a better way to specify things. What's lacking in the metadata is the information how a certain measurement is

derived does the timestamp 00 refer to the period 00-30 minutes or 30-00 minutes and has the value been obtained by averaging or as instantaneous value? The timestamp in the data files need to be ISO 8601 compliant (e.g. 2018-01-01T12:00:00Z) so that also the time-zone information is contained. The site metadata should also contain the standard deviations for BD, SOC, and lattice water (which is btw. completely missing). For the easting and northing the projection needs to be specified. Please provide also the hourly raw neutron counts (uncorrected) as well as the Snow-Fox raw and corrected counts as some people may be interested in using their own corrections.

AR32 Thank you for these suggestions. We will update the metadata to improve clarity and data accessibility for users, and enable easier download of files, which we agree will make the data set more usable. We currently provide hourly corrected neutron counts and will consider including the raw neutron counts in future uploads.

RC33 387: Fig. 10: Change to color-blind safe palette (https://knightlab.northwestern.edu/2016/07/18/three-tools-to-help-you-makecolorblind-friendly-graphics/)

AR33 The colours in Figure 10 will be amended.

RC34 435-447: It would be nice to have also a short outlook. What are the perspectives for COSMOS-UK? Are there plans to extend the network (e.g., for Scotland)? How long is the projected lifetime of the project. How will the CRNS sensors degrade over time? Do you plan to upgrade the network with more sensitive detectors as they become available on the market?

AR34 Thank you for your interesting questions. COSMOS-UK has been designed as a long-term monitoring network and aims to provide easily-accessible soil moisture and hydrometeorological data; this will continue to be the network's goal whilst dependent on funding. Regarding CRNS sensor degradation, we will consider performing more repeat calibrations to understand any changes. We will continue to consider sensor upgrades when available and feasible. We will amend the manuscript to reflect these

points.

References

Blonquist, J. M., Jones, S. B. and Robinson, D. A.: A time domain transmission sensor with TDR performance characteristics, J. Hydrol., 314(1–4), 235–245, doi:10.1016/j.jhydrol.2005.04.005, 2005.

Colli, M., Pollock, M., Stagnaro, M., Lanza, L. G., Dutton, M. and O'Connell, E.: A Computational Fluid-Dynamics Assessment of the Improved Performance of Aerodynamic Rain Gauges, Water Resour. Res., 54(2), 779–796, doi:10.1002/2017WR020549, 2018.

Köhli, M., Schrön, M., Zreda, M., Schmidt, U., Dietrich, P. and Zacharias, S.: Footprint characteristics revised for field-scale soil moisture monitoring with cosmic-ray neutrons, Water Resour. Res., 51(7), 5772–5790, doi:10.1002/2015WR017169, 2015.

Schrön, M., Köhli, M., Scheiffele, L., Iwema, J., Bogena, H. R., Lv, L., Martini, E., Baroni, G., Rosolem, R., Weimar, J., Mai, J., Cuntz, M., Rebmann, C., Oswald, S. E., Dietrich, P., Schmidt, U. and Zacharias, S.: Improving calibration and validation of cosmic-ray neutron sensors in the light of spatial sensitivity, Hydrol. Earth Syst. Sci., 21(10), 5009–5030, doi:10.5194/hess-21-5009-2017, 2017.

Seneviratne, S. I., Corti, T., Davin, E. L., Hirschi, M., Jaeger, E. B., Lehner, I., Orlowsky, B. and Teuling, A. J.: Investigating soil moisture-climate interactions in a changing climate: A review, Earth-Science Rev., 99(3–4), 125–161, doi:10.1016/j.earscirev.2010.02.004, 2010.

Strangeways, I.: Improving precipitation measurement, Int. J. Climatol., 24(11), 1443–1460, doi:10.1002/joc.1075, 2004.

Response to Referee 2

RC1 1. The main issue is with an absence of calibration details : Table 5 and the dis-

cussion preceding it provide elaborate details of the sampling procedure and determination of soil moisture in the lab. The effectiveness of calibration is however mentioned in just one sentence, "There was < 0.03 cm3 cm-3 difference in VWC between the soil moisture determined from these samples and the corresponding daily VWC value derived using the site's initial calibration data". Table 5 only shows the reference values and not the difference between the point scale measurements and the CRNS data. In the absence of such results, it is difficult to judge how good has the calibration been. Perhaps in the supplementary material, more detailed results of calibration could be given.

AR1 We will include additional information in Table 5 and amend the text to increase clarity. In our experience the point-scale sensor measurements on the day of calibration are unreliable due to poor contact between the sensor and the soil. This contact improves over time and qualitative comparisons have been made between the CRNS VWC and point measurements. Undertaking quantitative comparisons is planned.

RC2 2. The claim that the spatial data on soil moisture leads to improved hydrometeorological forecasts needs to be substantiated, either by citing appropriate references or through a convincing argument.

AR2 We will amend the text and add references.

RC3 3. Table 1 gives details of the COSMOS-UK stations. Please consider highlighting the mountainous sites, if any among these. The altitude values shown in the table do indicate the heights at which the stations are located, but a higher altitude does not necessarily indicate a station on a mountain. It would also be interesting to see the soil moisture signatures of the mountainous regions as compared to those in the plains.

AR3 The COSMOS-UK sites are installed in non-mountainous and largely flat locations due to the logistical challenges these regions present. We will make this clearer in the manuscript.

RC4 4. Is there any irrigation in the area around the CRNS stations? If yes, how has it affected the soil moisture data? Is it possible to filter out the effect of irrigation in the data?

AR4 Sites with regular irrigation were avoided when identifying locations for COSMOS-UK installations. We will be notified of any irrigation at the sites and will explore the impact on data if this occurs.

RC5 5. Table 4 shows that at the same depth two TDT point source sensors are located. How are they spaced? How is the data from the two sensors merged?

AR5 All TDT sensors at COSMOS-UK sites are installed in pairs. Each site has one pair of sensors buried at a depth of 10 cm, located 1 m apart. The additional array of 4 pairs of TDT sensors (where available) are buried at 5, 15, 25 and 50 cm depths, with each sensor located 30 cm from its paired sensor. A horizontal distance of 15 cm is ensured before burying the next pair of sensors at the desired depth. Data from each sensor are provided separately, and the user can choose whether to combine data across pairs. We will make this clearer in the manuscript.

RC6 6. Lines 183-184 : Volumetric samples are taken at five depths, upto 0.25 m bgl. However, the CRNS data is between 0.1 m to 0.8 m. Is there any calibration for depths below 0.25 m?

AR6 There is currently no adjusted calibration applied for depths below 0.25 m. Soil sampling depths were selected to match typical (moist) UK conditions. Ideally sites should have repeated calibrations at different VWCs to capture the full range of measured VWC. We will amend the text and emphasise that the shallow soil layers are given higher weighting than deeper layers to accommodate greater water contribution at shallow depths (Köhli et al., 2015; Schrön et al., 2017).

RC7 7. Fig 2 (appearing around line 272) should be Fig. 6.

AR7 Thank you for your comment. This is correct in the preprint version of the

manuscript.

RC8 8. It is difficult to understand the "automatic processing" for quality checks. Table 6 provides the flags raised for various events, but how these events are identified in the data is not clear (for example, how does the automatic processing detect simultaneously, missing data and small sample of data?)

AR8 The explanation will be amended to increase clarity.

RC9 9. Fig. 8 caption may be made more descriptive. Also, the caption states that these are the observations required for PE calculation, but the last panel in fact shows the PE calculated! It would also be interesting to see the soil temperature plot along with the other variables shown here (although soil temperature is not used in PE calculation).

AR9 Thank you for your comment and suggestion. We will improve the caption for Figure 8.

RC10 10. Please also discuss how the soil moisture measured at these 51 stations may be smoothened to upscale it to the national scale (see line no. 396).

AR10 We will add this discussion to the manuscript.

References

Köhli, M., Schrön, M., Zreda, M., Schmidt, U., Dietrich, P. and Zacharias, S.: Footprint characteristics revised for field-scale soil moisture monitoring with cosmic-ray neutrons, Water Resour. Res., 51(7), 5772–5790, doi:10.1002/2015WR017169, 2015.

Schrön, M., Köhli, M., Scheiffele, L., Iwema, J., Bogena, H. R., Lv, L., Martini, E., Baroni, G., Rosolem, R., Weimar, J., Mai, J., Cuntz, M., Rebmann, C., Oswald, S. E., Dietrich, P., Schmidt, U. and Zacharias, S.: Improving calibration and validation of cosmic-ray neutron sensors in the light of spatial sensitivity, Hydrol. Earth Syst. Sci., 21(10), 5009–5030, doi:10.5194/hess-21-5009-2017, 2017.

---

## Author Response (AR1)

**Response to Referees**

The authors thank both referees for providing detailed and helpful comments, suggestions and questions. We have added detail and clarity to the manuscript where suggested.

With regard to comments concerning the inclusion of individual calculations and more detailed explanations of variable derivation and methodology, we believe the manuscript describes the data set at an appropriate level of detail for the intended end users who are interested in soil moisture and hydrometeorology measurements. This description paper is not aimed at researchers looking to further develop the cosmic-ray method of soil moisture measurement.

The COSMOS-UK team is in active discussion with researchers across the European COSMOS community, with the intention of producing a more detailed output to describe specific methodologies for deriving soil moisture from the CRNS.

We address each referee comment (RC) with our response (AR) below.

**Response to Referee 1**

RC1

1: (Title) "empowering UK environmental science" To me this formulation reads a bit "selfish". It may be UK-national soil moisture data, but it should be (and I belief it is) open to anyone who works in the field of environmental science, British or not. I think there are many global or continental models that include the UK and would likewise benefit from your data. So I suggest to change the title into something less "nationalistic".

AR1

We had no intention for the title to imply any such meaning and we have therefore carefully amended it. The revised title is, "COSMOS-UK: National soil moisture and hydrometeorology data for environmental science research".

RC2

35: Could you find another citation Moene and van Dam? I think there are plenty of papers out there that give a more specific introduction to the topic of soil moisture in the Earth System, e.g., by Seneviratne et al.

AR2

We have referenced the work by Seneviratne et al. (2010).

RC3

40: "measurement footprint" You may already specify it here to let the reader know what to expect. E.g. "the field scale measurement footprint"

AR3

We have amended the text to "field scale measurement footprint".

RC4

57: 80cm -> blank missing

AR4

This has been corrected.

RC5

80-85: What's the reason for not having more stations in Scotland (only 2 stations in the east)? Also for Yorkshire and North-West England the network is more sparse.

AR5

Installation of COSMOS-UK sites was initially focussed in areas where soil moisture was expected to have greater variability and where practical and logistical challenges related to access and communication were minimised. Installation of sites in less-represented regions is in consideration but is dependent on the availability of resources. This is now clearer in the manuscript.

RC6

Table 1: I think the table needs to be condensed. Since available in the metadata, you can skip the 2 columns for Easting and Northing and also End date can be moved to Start date like this: "Start(end) date" 26/11/2013(-01/10/2016)

The numbers for SAAR and Altitude should be right-adjusted. For the Soil type and land cover you should define abbreviations in the table header (e.g. MS for Mineral soil, IG for Improved grassland). and Soil type should go right of Altitude and SAAR. So finally one line could be a s short as: Cochno 23/08/2017 168 662 MS IG and all stations could be overviewed at a glimpse one a single page.

AR6

Table 1 has been amended with these suggestions.

RC7

120-122: not clear if all CRS1000B sensors have been removed from the network and CRS2000Bs are used now everywhere. If all had been changed, since when is the network pure CRS 2000B? Pls. be more specific.

AR7

The text has been revised for clarity. Only the first four sites were installed with a CRS1000B (Chimney Meadows, Sheepdrove, Waddesdon and Wytham Woods), whilst all other sites were installed with a CRS2000B sensor. Wytham Woods was decommissioned in October 2016. In February 2020, CRS2000B sensors were installed at Chimney Meadows, Sheepdrove and Waddesdon alongside the existing CRS1000B sensors which remain connected.

RC8

125: Shortly explain what TDT sensors are.

AR8

An explanation has been added.

RC9

127: What was the reason to remove the PICO profiles?

AR9

We experienced issues with this sensor as configured at our sites. Specifically the configuration resulted in a high failure rate for other sensors and resulted in loss of data. Because of this, these instruments were removed during 2019 - 2020 in order to maximise the data capture across the array of instruments. This has been made clearer in the manuscript.

RC10

139: Why don't you use wind shields for the Pluvios? It's so often windy in Britain.

AR10

Pluvio wind protection shields were omitted during COSMOS-UK site installations as site locations were identified as being not particularly exposed. It is acknowledged that this decision could impact the extent of wind-induced under-catch, and shields will be considered during future network improvements. This is now clear in the manuscript.

RC11

157: Pls. list the 8 sites with extra snow measurements

AR11

The list has been added.

RC12

169: Is there any publication that shows the superiority of the SBS500, so that you want to use it as reference to check the quality of others (like the Pluvio)?

AR12

We have amended the text to include relevant citations. The design of the EML SBS500 has improved aerodynamic characteristics (Strangeways, 2004), and Colli et al. (2018) report that when compared with other assessed rain gauges, the SBS500 exhibits superior reduction in turbulence and under-catch.

RC13

183-184: Why do you only consider 0-25 cm depth for calibration? Later in figure 7 you suggest much larger penetration depths.

AR13

Soil sampling depths were selected to match typical (moist) UK conditions. Ideally, subject to cost, calibrations should be repeated at different VWCs, preferably capturing the full range of VWC measured at a particular site. Clarity has been added to the text and we have highlighted that the shallow soil layers are given higher weighting than deeper layers to represent the decreasing contribution of deeper soil water (Köhli et al., 2015; Schrön et al., 2017). Figure 7 has been amended to display the correct (lower) values for D86. We apologise for this error; the data set has been corrected.

RC14

183: Pls. provide model and manufacturer for the soil augers

AR14

We have included these details.

RC15

189: Fig.3: You should also provide annuluses (25,75, 200m) for the older calibrations (e.g. by having 2 separate graphs side by side with one legend, or by additional annotation in brackets '5 (25) m')

AR15

Figure 3 has been amended.

RC16

205: Were the bulk densities also obtained with the weighting function?

AR16

We do not currently obtain the bulk densities with the weighting function, however we will consider this in future work. This is now clearer in the manuscript.

RC17

206: Did you also consider the findings of Schroen et al. 2017 https://hess.copernicus.org/articles/21/5009/2017/ for your weighting and calibration approach? The Sheepdrove data of COSMOS-UK was used in this work.

AR17

Yes these findings have been considered in our approach; when applying the soil sampling calibration, the shallow 5 cm layers are given a higher weighting than the deeper ones. This is now clearer in the manuscript. Please note that the Sheepdrove COSMOS-UK site is unrelated and in a different location to the Sheepdrove monitoring station referred to by Schrön et al. (2017).

RC18

208-210: There should be a discussion about the magnitude of possible errors in the determination of the reference soil moisture. Is the 0.03 vol.% difference significant or may the error in sampling and thermo-gravimetry be even higher?

AR18

Detail has been added to the manuscript and we plan to address this as part of future recalibrations.

RC19

211: Tab. 5: Please give standard deviations for bulk density, lattice water, and soil organic carbon (add with ± sign)

AR19

These have been added to the data set.

RC20

217-230: Please be more specific in the description of the derivation of the corrected neutron count signal. This important step should be reproducible by the data users. So you should provide the exact

formulas and specify the constants used (e.g. for the reference pressure). Did you use the same reference pressure for all stations? Did you do or do you plan to do a cross-calibration for the different sites, to get an idea of the variations in sensitivity of the sensors?

AR20

After consideration we believe that the current description with relevant citations provides an acceptable level of detail for the data set's intended users, who are interested in soil moisture and other environmental data.

We have included additional detail regarding reference pressure and CRNS cross-comparisons. A value of 1000 hPa is used for all sites. A selection of CRS2000/B sensors deployed at COSMOS-UK sites were cross-compared for sensitivity under lab conditions prior to installation. Where necessary, in situ sensitivity comparisons have been completed by collecting several months of data from adjacent sensors.

RC21

224: I think it should be "Physikalisches Institut's, University of Bern, Jungfraujoch"

AR21

The text has been corrected.

RC22

227: What are the implications of using a neutron monitor with such a large displacement and elevation difference. I assume a big difference in the cutoff rigidity between Jungfraujoch and your sites. Wouldn't the Kiel monitor be better suited? I think that this should also be discussed in the paper.

AR22

We have added detail to the manuscript. We use normalised count rates in the intensity correction, which are not greatly affected by their cut-off rigidity in the absence of significant space weather events. During such events, the cut-off rigidity of a specific location may change due to magnetic field disturbances, so only matching cut-off rigidity might not compensate for such events. When comparing Jungfraujoch neutron monitor with the available monitors of similar cut-off rigidities to the COSMOS-UK sites, the normalised counts and the associated trends were in good agreement. When choosing the most suitable neutron monitor for this work, Jungfraujoch was identified as a well-maintained monitor with a high level of data completeness, however we will continue to monitor research in this area to improve our methods where possible. The Kiel monitor is currently described as not having efficiency-corrected values available, so we will reconsider this monitor when these data are available.

RC23

241: Are the fits for all the sites performing equally well? You may discuss and add some other cases too.

AR23

There may be some misunderstanding with this question. This figure does not illustrate goodness of fit, it only illustrates the portion of the calibration curve corresponding to the observed count rates. We have made this clearer in the manuscript.

RC24

243: Fig. 4: Please add the formula for the synthetic curve including constants to the legend of the plot

AR24

The formula has been included in the text.

RC25

264: Fig. 5: Please add panel IDs (a, b, c). What is the reason for the counts in a) being half of those in b)? Is it a different aggregation interval? X-axes text (date) should be removed for all but the lowermost panel. Then the margins between panels can be removed in favor of larger legend font sizes. Y-axis text for e) Precipitation is too small (smaller than for other panels). Legends (font sizes) for d) VWC are too small.

AR25

Panel IDs have been added to Figure 5. Data have been replaced with the intended aggregation interval. The figure has also been updated to reflect the other suggestions.

RC26

271: Fig. 6: Please add panel IDs. Resolution of figure needs to be improved. Since colors are rather hard to distinguish (especially with red-green blindness), I would recommend to use gray-scale (maybe with transparency) for the lines and symbols (point, cross, dot, ...) for the regions at the highlighted date.

AR26

Figure 6 has been updated.

RC27

289: "measurements" I wouldn't see this as a measurement but rather as a derived quantity. Something of "shows the assumed/computed/estimated D86..."

AR27

The text has been amended.

RC28

293: Fig. 7: Add panel IDs. All texts are by far too small. As for Fig. 5 I suggest to remove the x-axes annotation and to cut the margin between the sub-panels. How can the Euston VWC time-series reach 0? Was that reflected by the TDT sensors?

AR28

Figure 7 has been amended. The Euston site is located on very well-draining sandy soil and during the extreme dry event in 2018 the derived soil moisture data were consistently low. The minimum daily average TDT VWC readings between 2018-05-01 and 2018-08-31 were: 1.3, 0.7, 2.3, 0.5, 2.7, 1.6, 2.2, 0.2, 0.3 and 0.4 at the respective depths of 5, 5, 10, 10, 15, 15, 25, 25, 50 and 50 cm.

RC29

328: As you provided SnowFox derived SWE in the data set, its derivation should be documented more precisely (formula, calibration). At least provide it as supplementary material.

AR29

The data set only provides SWE derived from the CRNS (not the SnowFox). We have made this clearer in the manuscript.

RC30

334: Fig. 8: Same as before (panel IDs and description in the caption), remove x-axes between sub-panels, increase font sizes and avoid red-green coloring in one graph.

AR30

Figure 8 has been amended.

RC31

353: Please explain what "gauge boards" are

AR31

We have amended the text.

RC32

361-386: The different data sub-products are clearly distinguished and organized in a logical structure. However, on top-level, the user doesn't want to find a 200 files long list to be downloadable click by click. So all the data should be organized as a single file archive (zip) download.

The user should be able to use the data set without considering the data-description paper just by making use of the metadata.

Thus, the information in table 7 should be provided within 4 additional metadata files (SH, SH...QC, Hourly, Daily). And the JSON format might be a better way to specify things. What's lacking in the metadata is the information how a certain measurement is derived does the timestamp 00 refer to the period 00-30 minutes or 30-00 minutes and has the value been obtained by averaging or as instantaneous value?

The timestamp in the data files need to be ISO 8601 compliant (e.g. 2018-01-01T12:00:00Z) so that also the time-zone information is contained.

The site metadata should also contain the standard deviations for BD, SOC, and lattice water (which is btw. completely missing). For the easting and northing the projection needs to be specified.

Please provide also the hourly raw neutron counts (uncorrected) as well as the Snow-Fox raw and corrected counts as some people may be interested in using their own corrections.

AR32

Access to the data has been improved with the ability to download zip files. The data set has been updated to include the suggested metadata.

RC33

387: Fig. 10: Change to color-blind safe palette (https://knightlab.northwestern.edu/2016/07/18/three-tools-to-help-you-makecolorblind-friendly-graphics/)

AR33

The colours in Figure 10 have been amended.

RC34

435-447: It would be nice to have also a short outlook. What are the perspectives for COSMOS-UK? Are there plans to extend the network (e.g., for Scotland)? How long is the projected lifetime of the project. How will the CRNS sensors degrade over time? Do you plan to upgrade the network with more sensitive detectors as they become available on the market?

AR34

We have amended the manuscript to reflect these points. COSMOS-UK has been designed as a long-term monitoring network and aims to provide easily-accessible soil moisture and hydrometeorological data; this will continue to be the network's goal whilst dependent on funding. Regarding CRNS sensor degradation, we will consider performing more repeat calibrations to understand any changes. We will continue to consider sensor upgrades when available and feasible.

References

Blonquist, J. M., Jones, S. B. and Robinson, D. A.: A time domain transmission sensor with TDR performance characteristics, J. Hydrol., 314(1–4), 235–245, doi:10.1016/j.jhydrol.2005.04.005, 2005.

Colli, M., Pollock, M., Stagnaro, M., Lanza, L. G., Dutton, M. and O'Connell, E.: A Computational Fluid-Dynamics Assessment of the Improved Performance of Aerodynamic Rain Gauges, Water Resour. Res., 54(2), 779–796, doi:10.1002/2017WR020549, 2018.

Köhli, M., Schrön, M., Zreda, M., Schmidt, U., Dietrich, P. and Zacharias, S.: Footprint characteristics revised for field-scale soil moisture monitoring with cosmic-ray neutrons, Water Resour. Res., 51(7), 5772–5790, doi:10.1002/2015WR017169, 2015.

Schrön, M., Köhli, M., Scheiffele, L., Iwema, J., Bogena, H. R., Lv, L., Martini, E., Baroni, G., Rosolem, R., Weimar, J., Mai, J., Cuntz, M., Rebmann, C., Oswald, S. E., Dietrich, P., Schmidt, U. and Zacharias, S.: Improving calibration and validation of cosmic-ray neutron sensors in the light of spatial sensitivity, Hydrol. Earth Syst. Sci., 21(10), 5009–5030, doi:10.5194/hess-21-5009-2017, 2017.

Seneviratne, S. I., Corti, T., Davin, E. L., Hirschi, M., Jaeger, E. B., Lehner, I., Orlowsky, B. and Teuling, A. J.: Investigating soil moisture-climate interactions in a changing climate: A review, Earth-Science Rev., 99(3–4), 125–161, doi:10.1016/j.earscirev.2010.02.004, 2010.

Strangeways, I.: Improving precipitation measurement, Int. J. Climatol., 24(11), 1443–1460, doi:10.1002/joc.1075, 2004.

**Response to Referee 2**

RC1

1. The main issue is with an absence of calibration details : Table 5 and the discussion preceding it provide elaborate details of the sampling procedure and determination of soil moisture in the lab. The effectiveness of calibration is however mentioned in just one sentence, "There was < 0.03 cm3 cm-3

difference in VWC between the soil moisture determined from these samples and the corresponding daily VWC value derived using the site's initial calibration data". Table 5 only shows the reference values and not the difference between the point scale measurements and the CRNS data. In the absence of such results, it is difficult to judge how good has the calibration been. Perhaps in the supplementary material, more detailed results of calibration could be given.

AR1

The manuscript has been amended to increase clarity regarding calibration. In our experience the point-scale sensor measurements on the day of calibration, generally occurring shortly after installation, are unreliable due to poor contact between the sensor and the soil. This contact improves over time and qualitative comparisons have been made between the CRNS VWC and point measurements. Undertaking quantitative comparisons is planned.

RC2

2. The claim that the spatial data on soil moisture leads to improved hydrometeorological forecasts needs to be substantiated, either by citing appropriate references or through a convincing argument.

AR2

We have amended the text.

RC3

3. Table 1 gives details of the COSMOS-UK stations. Please consider highlighting the mountainous sites, if any among these. The altitude values shown in the table do indicate the heights at which the stations are located, but a higher altitude does not necessarily indicate a station on a mountain. It would also be interesting to see the soil moisture signatures of the mountainous regions as compared to those in the plains.

AR3

The COSMOS-UK sites are installed in non-mountainous and largely flat locations due to the logistical challenges these regions present. We have made this clearer in the manuscript.

RC4

4. Is there any irrigation in the area around the CRNS stations? If yes, how has it affected the soil moisture data? Is it possible to filter out the effect of irrigation in the data?

AR4

We have amended the manuscript. Sites with regular irrigation were avoided when identifying locations for COSMOS-UK installations. We will be notified of any irrigation at the sites and will explore the impact on data if this occurs.

RC5

5. Table 4 shows that at the same depth two TDT point source sensors are located. How are they spaced? How is the data from the two sensors merged?

AR5

All TDT sensors at COSMOS-UK sites are installed in pairs. Each site has one pair of sensors buried at a depth of 10 cm, located 1 m apart. The additional array of 4 pairs of TDT sensors (where available) are

buried at 5, 15, 25 and 50 cm depths, with each sensor located 30 cm from its paired sensor. A horizontal distance of 15 cm is ensured before burying the next pair of sensors at the desired depth. Data from each sensor are provided separately, and the user can choose whether to combine data across pairs. We have made this clearer in the manuscript.

RC6

6. Lines 183-184 : Volumetric samples are taken at five depths, upto 0.25 m bgl. However, the CRNS data is between 0.1 m to 0.8 m. Is there any calibration for depths below 0.25 m?

AR6

There is currently no adjusted calibration applied for depths below 0.25 m. Soil sampling depths were selected to match typical (moist) UK conditions. Ideally sites should have repeated calibrations at different VWCs to capture the full range of measured VWC. We have amended the manuscript and emphasised that the shallow soil layers are given higher weighting than deeper layers to accommodate greater water contribution at shallow depths (Köhli et al., 2015; Schrön et al., 2017).

RC7

7. Fig 2 (appearing around line 272) should be Fig. 6.

AR7

This is correct in the manuscript.

RC8

8. It is difficult to understand the "automatic processing" for quality checks. Table 6 provides the flags raised for various events, but how these events are identified in the data is not clear (for example, how does the automatic processing detect simultaneously, missing data and small sample of data?)

AR8

The explanation has been amended to increase clarity.

RC9

9. Fig. 8 caption may be made more descriptive. Also, the caption states that these are the observations required for PE calculation, but the last panel in fact shows the PE calculated! It would also be interesting to see the soil temperature plot along with the other variables shown here (although soil temperature is not used in PE calculation).

AR9

We have improved the caption for Figure 8.

RC10

10. Please also discuss how the soil moisture measured at these 51 stations may be smoothened to upscale it to the national scale (see line no. 396).

AR10

We have amended the manuscript.

References

Köhli, M., Schrön, M., Zreda, M., Schmidt, U., Dietrich, P. and Zacharias, S.: Footprint characteristics revised for field-scale soil moisture monitoring with cosmic-ray neutrons, Water Resour. Res., 51(7), 5772–5790, doi:10.1002/2015WR017169, 2015.

Schrön, M., Köhli, M., Scheiffele, L., Iwema, J., Bogena, H. R., Lv, L., Martini, E., Baroni, G., Rosolem, R., Weimar, J., Mai, J., Cuntz, M., Rebmann, C., Oswald, S. E., Dietrich, P., Schmidt, U. and Zacharias, S.: Improving calibration and validation of cosmic-ray neutron sensors in the light of spatial sensitivity, Hydrol. Earth Syst. Sci., 21(10), 5009–5030, doi:10.5194/hess-21-5009-2017, 2017.